New species of Ontocetus (Pinnipedia: Odobenidae) from the Lower Pleistocene of the North Atlantic shows similar feeding adaptation independent to the extant walrus (Odobenus rosmarus)

Boisville Mathieu 1 mathieu.boisville1@gmail.com
http://orcid.org/0000-0003-0449-8574 Chatar Narimane 2 3
http://orcid.org/0000-0001-5329-4063 Kohno Naoki 1 4
1 Earth Historical Analysis, Earth Evolution Sciences, Graduate School of Life and Environmental Sciences, University of Tsukuba , Tsukuba, Ibaraki , Japan
2 Evolution & Diversity Dynamics Lab, Department of Geology, University of Liège , Liège , Belgium
3 Functional Anatomy and Vertebrate Evolution Lab, Department of Integrative Biology, University of California Berkeley , Berkeley, California , United States
4 Department of Geology and Paleontology, National Museum of Nature and Science , Tsukuba, Ibaraki , Japan
Moncunill-Solé Blanca
Electronic publication date: 2024 Aug 13
Publication date: 2024
Volume: 12
Electronic Location ID: e17666
Received 2023 Dec 27; Accepted 2024 Jun 11
Copyright: © 2024 Boisville et al.
Copyright year: 2024
Copyright holder: Boisville et al.
License: This is an open access article distributed under the terms of the Creative Commons Attribution License, which permits unrestricted use, distribution, reproduction and adaptation in any medium and for any purpose provided that it is properly attributed. For attribution, the original author(s), title, publication source (PeerJ) and either DOI or URL of the article must be cited.
License URL: https://creativecommons.org/licenses/by/4.0/

Keywords: Ontocetus, Pliocene, United Kingdom, Belgium, Odobenus, mandibles, Pleistocene, Suction-feeding

Funding: Hans de Bruijn Foundation KvK-nummer 30178293 Japan Society for the Promotion of Science 22H02687 University of Tsukuba with the Overseas Musha This work was supported by the Hans de Bruijn Foundation (KvK-nummer 30178293), Japan Society for the Promotion of Science (JSPS) (22H02687) and the University of Tsukuba with the Overseas Musha Training Support Program for graduate students. The funders had no role in study design, data collection and analysis, decision to publish, or preparation of the manuscript.

==============================
Ontocetus is one of the most notable extinct odobenines owing to its global distribution in the Northern Hemisphere. Originating in the Late Miocene of the western North Pacific, this lineage quickly spread to the Atlantic Ocean during the Pliocene, with notable occurrences in England, Belgium, The Netherlands, Morocco and the eastern seaboard of the United States. Reassessment of a pair of mandibles from the Lower Pleistocene of Norwich (United Kingdom) and a mandible from the Upper Pliocene of Antwerp (Belgium) that were referred to as Ontocetus emmonsi reveals existences of features of both Ontocetus and Odobenus. The presence of four post-canine teeth, a lower canine larger than the cheek-teeth and a lower incisor confirms the assignment to Ontocetus; simultaneously, characteristics such as a fused and short mandibular symphysis, a well-curved mandibular arch and thin septa between teeth align with traits usually found in Odobenus. Based on a combination of these characters, we describe Ontocetus posti, sp. nov. Its mandibular anatomy suggests, a better adaptation to suction-feeding than what was previously described in the genus suggesting that Ontocetus posti sp. nov. likely occupied a similar ecological niche to the extant walrus Odobenus rosmarus. Originating from the North Pacific Ocean, Ontocetus most likely dispersed via the Central American Seaway. Although initially discovered in the Lower Pliocene deposits of the western North Atlantic, Ontocetus also left its imprint in the North Sea basin and Moroccan Plio-Pleistocene deposits. The closure of the Isthmus of Panama during the Mio-Pliocene boundary significantly impacted the contemporary climate, inducing global cooling. This event constrained Ontocetus posti in the North Sea basin leaving the taxon unable to endure the abrupt climate changes of the Early Pleistocene, ultimately going extinct before the arrival of the extant counterpart, Odobenus rosmarus.

Introduction

The walrus (Odobenus rosmarus) stands as one of the most iconic mammals of the Arctic, distinguished from all other pinnipeds by its immense size and prominent tusks (Fay, 1982). Walruses are one of the largest extant members of Carnivora, males of this species weighing up to 2.5 tons with an average length of 3 m. Walruses inhabit shallow, cold Arctic waters and males form small harems during the breeding season (Wiig et al., 2007).

Their primary diet consists of bivalve mollusks such as clams (Ruditapes decussatus or Arctica islandica) that they capture using a unique “suction-feeding” technique, utilizing lips, tongue, and arched palate as a piston to extract mollusk flesh (Fay, 1982; Kastelein & Gerrits, 1990; Kastelein, Gerrits & Dubbeldam, 1991). Emblematic anatomical characteristics of this feeding adaptation include the displacement of the upper incisors that have shifted laterally and adjacent to the upper canines, while the lower incisors have disappeared (Montague Cobb, 1933).

As the sole extant member of the family Odobenidae, walruses represent a lineage with notable diversity, particularly in the fossil record within the pinnipeds (Boessenecker & Churchill, 2021). While extant walruses predominantly inhabit the Arctic, their extinct relatives once occupied temperate and subtropical latitudes, mainly in the Miocene eastern North Pacific realm, with a wide range of shape and size, from small sea lion-like ‘imagotariines’ to bizarre unique double-tusked dusignathines (Repenning & Tedford, 1977; Deméré, 1994a, 1994b; Kohno, 1994; Miyazaki et al., 1995; Boessenecker, 2013, 2017; Magallanes et al., 2018; Boessenecker & Churchill, 2021).

Originating in the North Pacific, tusked walruses (Odobenini sensu Deméré, 1994b) later dispersed to the North Atlantic, likely during Miocene-Pliocene transition (Barnes & Perry, 1989; Horikawa, 1995; Kohno et al., 1995; Kohno, Narita & Koike, 1998; Boessenecker, 2017). The Odobenini comprises several fossil taxa, including poorly known ones like Pliopedia pacifica from the Upper Miocene-Lower Pliocene deposits of California (Kellogg, 1921; Repenning & Tedford, 1977; Boessenecker et al., 2024) and Protodobenus japonicus from Lower Pliocene formations in Japan (Horikawa, 1995; Magallanes et al., 2018), along with the highly specialized suction-feeder Valenictus Mitchell, 1961 from the Upper Miocene-Upper Pliocene of California (Deméré, 1994a; Boessenecker, 2013, 2017; Boessenecker et al., 2024).

Within the Odobenini, Ontocetus is the tusked walrus with the richest fossil record and latitudinally-wide distribution in the Northern Hemisphere (Boessenecker & Churchill, 2021). Over nearly a century and a half, Ontocetus has been a subject of controversy (Kohno & Ray, 2008). With its oldest fossil record tracked back to the Miocene/Pliocene boundary age of Japan (Okamoto & Kohno, 2019), indicating a North Pacific origin, Ontocetus dispersed around the Mio-Pliocene boundary, to the Atlantic Ocean through the Central American Seaway (Repenning, Ray & Grigorescu, 1979; Kohno et al., 1995; Kohno, Narita & Koike, 1998; Kohno & Ray, 2008), although a dispersal via the Arctic was also proposed by various authors (Gladenkov et al., 2002; Deméré, Berta & Adam, 2003; Boessenecker, Boessenecker & Geisler, 2018; Boessenecker & Churchill, 2021). A large amount of specimens have been discovered in younger deposits along the east coast of the United States (e.g., Leidy, 1859; Berry & Gregory, 1906; Ray, 1960; Morgan, 1994; Kohno & Ray, 2008; Boessenecker, Boessenecker & Geisler, 2018), in the North Sea basin such as England (e.g., Lankester, 1880, 1882; Newton, 1882, 1891), Belgium (e.g., du Bus, 1867; van Beneden, 1871, 1877; Hasse, 1909, 1911; Rutten, 1907; Misonne, 1958) and The Netherlands (e.g., van Deinse, 1964; van der Feen, 1968; Erdbrink & van Bree, 1990, 1999a, 1999b, 1999c) but also in Morocco (Geraads, 1997), mainly represented by isolated upper canines although a few dozen skulls, humeri, and mandibles have been discovered all over the Northern Hemisphere. In the past, a total of five genera and eight species of Plio-Pleistocene Ontocetus-like walruses from the North Atlantic and North Sea have been described: Ontocetus emmonsi Leidy, 1859, Trichecodon huxleyi Lankester, 1865, Alachtherium cretsii du Bus, 1867, Trichecodon koninckii van Beneden, 1871, Prorosmarus alleni Berry & Gregory, 1906, Trichechus antverpiensis Rutten, 1907, A. antwerpiensis Hasse, 1909, and A. africanum Geraads, 1997. More recently, some specimens were discovered in Japan and described although no species name was attributed to those remains which includes tusks, skulls and appendicular elements (Yasuno, 1988; Kohno et al., 1995, Kohno, Narita & Koike, 1998; Okamoto & Kohno, 2019). A key challenge has been the definition of these species based on disparate skeletal elements by the different authors (e.g., crania, mandibles, humeri or tusks) (van Beneden, 1877; Berry & Gregory, 1906; Hasse, 1911; Ray, 1960, 1975; Erdbrink & van Bree, 1986, 1990; Post, 2004; Boessenecker, Boessenecker & Geisler, 2018). A 2008 review by Kohno and Ray focused on North Atlantic fossil material from the Pliocene formations, concluded that all the specimens from the North Sea were thought to belong in the ontogenetic and sexual variation of Ontocetus emmonsi Leidy, 1859 from the east coast of North America.

Ontocetus did not coexist with the genus Odobenus at least in the North Atlantic (Boessenecker, Boessenecker & Geisler, 2018), and did not share a direct common ancestor (Deméré, 1994b; Magallanes et al., 2018). Odobenus arrived in the North Atlantic not earlier than 244 Ka according to fossil occurrences (Sanders, 2002), while the last population of Ontocetus became extinct during Early Pleistocene (1.8–1.1 Ma) based on the record of the fossil from South Carolina, USA (Boessenecker, Boessenecker & Geisler, 2018). The factors behind the disappearance of Ontocetus from the North Atlantic remain poorly understood, but given the temporal separation from Odobenus, competition between these two genera can be excluded.

The present study focuses on re-evaluating a pair of mandibles from the Lower Pleistocene of Norwich, England, and a mandible from the Upper Pliocene of Antwerp, Belgium. At least the latter was historically reported under different taxonomic names (Trichecodon koninckii by van Beneden (1877); Odobenus koninckii by Deméré (1994b)), but these names have been discarded as nomen nodum or nomen dubium (e.g., Rutten, 1907; van der Feen, 1968; Kohno & Ray, 2008) because of quite fragmentally nature of the holotype (isolated broken tusk), which is survived only by a cast. Using these materials, we aim (1) to re-evaluate two newly discovered mandibles from the Upper Pliocene of Belgium and the Lower Pleistocene of the United Kingdom and The Netherlands; (2) to discuss the paleoecology of Ontocetus concerning the vacant niche in the absence of Odobenus; and (3) to contextualize the extinction of Ontocetus in relation to climate changes during the Early Pleistocene.

Materials and Methods

Systematic paleontology

The original specimen (NWHCM 1996.1) was photographed by the last author on June 29, 2000 at the Norwich Castle Museum (NWHCM). The cast (NMR 7472) was photographed by the senior author on March 3, 2020 at the Institut Royal des Sciences Naturelles de Belgique (IRSNB). Both mandibles (NMR 7472, IRSNB M156) were taken via a Nikon D800E with AF-Nikkor 50 mm F1.8 lens and a Panasonic Lumix G1 with Olympus 14–42 mm lens. The figures were performed in Adobe Photoshop CS6 and Adobe Illustrator CS6. The anatomical terminology used in the present study follows Deméré (1994a, 1994b), Kryukova (2012), Boessenecker & Churchill (2013), Velez-Juarbe (2017) and Magallanes et al. (2018) for the walrus and Evans & de Lahunta (2013) for the domestic dog, as a representative for the (caniform) Carnivora. Terminologies for the mandibular teeth have been abbreviated (incisors = i, canine = c, premolars = p, first premolar = p1, second premolar = p2, third premolar = p3, fourth premolar = p4, molars = m). Terminologies for the musculature and inference of muscle insertions (areas) are inspired by Valentin (1990), Kastelein, Gerrits & Dubbeldam (1991), Lavergne, Vanneuville & Santoni (1996), Naples, Martin & Babiarz (2011) and Tseng et al. (2011).

This study includes the following morphological measurements: the minimum mandible thickness (MT) and the least mandible depth (MD), following Wiig et al. (2007). The MT is the minimum transverse thickness of the mandible posterior to the last postcanine; the MD is the minimal vertical height of the mandible posterior to the last post-canine. MT and MD have been measured using a digital caliper with an accuracy of 0.01 mm. Five angular measurements have also been taken, following Mohr (1942) and Deméré (1994a) and showing in Boisville et al. (2022): angle between (a) the anterior and dorsal margin; (b) the anterior and ventral margins; (c) the ventral and dorsal margins; (d) the medial edge of the condyle and the mandibular symphysis; (e) the lateral edge of the mandibular terminus and the symphysis; f) the horizontal and vertical rami; and (g) the coronoid process and the mandibular condyle. Other mandibular and tooth measurements follow Erdbrink & van Bree (1986, 1990), Deméré (1994b), Kohno & Ray (2008) and Magallanes et al. (2018). All measurements are provided as Table S1.

Morphometry

Multiple measurements were taken capturing the overall shape and proportions of the lower dentition and mandible (a total of 31 linear measurements and seven angles) to compare the new species to Ontocetus emmonsi and Odobenus rosmarus. All morphometric analyses were run using the R statistical environment v 4.2.3 (R Development Core Team, 2023). The R script used to run the analysis is provided in the Supplemental Data 1. We applied a 50% completeness threshold, the female Ontocetus posti (RGM.St.119589) did not pass the threshold as well as two specimens of Ontocetus emmonsi (female USNM PAL 374273 and male ROM 26116) due to their preservation stage. The morphological dataset was imported using the ‘read.csv’ function and then scaled (z-transform), distance matrices (based on pairwise dissimilarities) were computed from these scaled datasets using the ‘dist’ function from the stats v4.3.0 package. We generated morphospaces using: a Principal Coordinates Analysis (PCoA), using the ‘pcoa’ function implemented in the ape package (Paradis & Schliep, 2019) and a cluster using the hclust function from the stats v4.3.0 package. The PcoA retrieved 13 axes, the first two explaining a total of 65.36% of variance (43.02% and 22.34% respectively). Finally, a permutational multivariate analysis of variance (PerMANOVA, formerly known as non-parametric/NP-MANOVA) (Anderson, 2001) was performed using the ‘adonis2’ function from the vegan package (Dixon, 2003). PerMANOVA was performed (1,000 permutations using the ‘euclidean’ method) on the distance matrix of the ratios to test for significant differences between the new species and Ontocetus emmonsi, but also between the new species and Odobenus rosmarus.

Phylogeny

Our phylogenetic analyses are based on the morphological character matrix published by Boessenecker et al. (2024) including fossil and extant Odobenidae with various outgroup taxa were also selected from other pinniped families, including two extinct desmatophocids (Desmatophoca oregonensis Condon, 1906 and Allodesmus kernensis Kellogg, 1922), two early diverging “enaliarctines” (Enaliarctos emlongi Berta, 1991 and Pteronarctos goedertae Barnes & Perry, 1989), one otariid (Callorhinus ursinus), and two extant phocids (Erignathus barbatus Erxleben, 1777 and Monachus monachus Hermann, 1779). We used Mesquite 3.81 (Maddison & Maddison, 2023) to add the new species as a new OTU (Operational Taxonomic Unit), however, considering the material described we could only score a limited number of characters (characters 77–90, 94–95, 99–106, 122–129; see Files S1 and S2).

Some modifications were made from the matrix of Boessenecker et al. (2024). First, characters 78 and 82 were fused in a new character 78 describing both the mandibular symphysis and the potential presence of a furrow: 0 = absent, 1 = present with furrow, 2 = present without furrow. Character 81 was modified to 0 = same level or anterior to p2, 1 = posterior to level of p2 (the polymorphic state was removed and polymorphic specimens were coded as 0 and 1), same for character 83. The polymorphism as an independent state was excluded because sexual dimorphism may affect the coding of character states. Male character states are generally more definitive compared to those of female individuals, because of hypermorphotic nature in males. Therefore, if possible, using male specimens is preferable for coding characters. Characters 81, 83 (now 82), 99 (now 98), 127 (now 126) were treated as unordered in order to not exclude independent evolutionary changes and possible convergent evolution. We also included two new characters: Character 144 describing the p1 as follows (0 = present, 1 = absent) as well as character 145 describing the direction of coronoid process projection (0 = vertical, 1 = posterodorsally projected).

We used TNT v 1.5 (Goloboff & Catalano, 2016) to perform the phylogenetic analysis under implied weighting using a script modified from Chatar, Michaud & Fischer (2022). We expanded the memory of TNT to a maximum of 100,000 trees. We set the search parameters at: New Technology Search, 200 ratchet iterations, 10 cycles of drifting, five hits and five replications for each hit. We then used the tree branch bisection and reconnection algorithm (TBR) to fully explore the tree islands identified by the ratchet. Nodal support was measured using symmetric resampling with 1,000 replications, each replication involving a New Technology search with a change probability of 33%. We choose symmetric resampling over bootstrapping or jack-knifing, as this measure is not affected by character weighting and is thus more appropriate to deal with implied weights (Goloboff et al., 2003) (see File S2). The best score was obtained when K was set to 12 so we computed the time calibration on the strict consensus cladogram for K = 12. To do so, we used the ‘equal’ method of the ‘timePaleoPhy’ function from the strap v1.4 package in R (Bell & Lloyd, 2015) and a table containing the biostratigraphic range of each OTU obtained from Dewaele, Lambert & Louwye (2018a), Magallanes et al. (2018), Boessenecker et al. (2024). We then generated the time tree using the ‘geoscalePhylo’ function from the paleotree v3.3.25 package (Bapst, 2012).

Results

Systematic paleontology

Mammalia Linnaeus, 1758

Carnivora Bowdich, 1821

Pinnipedia Illiger, 1811

Odobenidae Allen, 1880

Odobeninae Mitchell, 1968

Odobenini Deméré, 1994b

Genus Ontocetus Leidy, 1859

Type species: Ontocetus emmonsi Leidy, 1859; Yorktown Formation, North Carolina, Pliocene

Amended diagnosis of genus

A genus of the subfamily Odobeninae distinguished from other genera by having tusk-like canine with thin cementum and orthodentine layers and trefoiled incisive foramina; distinguished from Aivukus and Protodobenus by having tusk-like upper canine with well-defined core of globular osteodentine; distinguished from Valenictus and Odobenus by retention of the tusk-like upper canine with strong curvature, taper, lateral compression, and longitudinal surface fluting; two upper incisors not in line with the cheek teeth; two well developed lower incisors; distinguished from Odobenus by having a slanted and elevated anterior margin of the horizontal ramus; a short distance between the dorsal margin of the horizontal ramus and coronoid process; a lower canine at least 20% larger than cheektooth row; the presence of p1 within lower postcanine toothrow; and a high and posterolaterally projected coronoid process.

Ontocetus posti sp. nov.

Trichecodon koninckii (in part) van Beneden, 1877

Trichecodon huxleyi (in part) Kellogg, 1922

Odobenus huxleyi (in part) Hooijer, 1957

Odobenus koninckii (in part) Deméré, 1994b

Ontocetus emmonsi (in part) Kohno & Ray, 2008

LSID

urn:lsid:zoobank.org:act:14595AD3-E3F5-461C-834E-DF275B95E80D

The electronic version of this article in Portable Document Format (PDF) will represent a published work according to the International Commission on Zoological Nomenclature (ICZN), and hence the new names contained in the electronic version are effectively published under that Code from the electronic edition alone. This published work and the nomenclatural acts it contains have been registered in ZooBank, the online registration system for the ICZN. The ZooBank LSIDs (Life Science Identifiers) can be resolved and the associated information viewed through any standard web browser by appending the LSID to the prefix http://zoobank.org/. The LSID for this publication is: urn:lsid:zoobank.org:pub:13F7B996-8009-47B7-A8B6-353D7D14ED81. The online version of this work is archived and available from the following digital repositories: PeerJ, PubMed Central SCIE and CLOCKSS.

Diagnosis of species

A species of Ontocetus that exhibits mosaic characters including some shared traits with the genus Odobenus. These diagnostic features of Ontocetus comprise a lower canine markedly larger (at least 20% larger than cheekteeth) from the cheek toothrow, presence of the lower first premolar (p1), specifically at least four lower postcanine teeth that are consisted of p1–p4, a broad and short edentulous mandibular terminus exhibiting two lower incisors (i2, i3), an elevated and slanted anterior margin, the relative size of the mandible (reaching almost 30 cm in horizontal length), a high, thin, and posterolaterally projected coronoid process, as well as a short distance between the dorsal margin and coronoid process (less than 8% of the total length of the mandible).

This species also shares certain characteristics with Odobenus, such as the fusion of the mandibular symphysis, a mandibular symphysis oval in shape and shorter in proportion to the total length of the mandible (less than 50%), a horizontal ramus with a well-curved lateral occlusal outline (also occurs in the referred mandible of Dusignathus seftoni) that accounts for the shortening of the rostrum. Additionally, a shorter space between the canine and the cheek teeth can be observed (less than 10% of the total length of the dental row), as well as between the thin cheek tooth alveoli themselves. Other shared traits encompass an underdeveloped angular process, and a flat, less developed mandibular condyle.

Etymology

The species is named in honor of Mr. Klaas Post, curator of Vertebrate Paleontology at Natuurhistorisch Museum Rotterdam, in recognition of his extensive contributions to geology and paleontology in the North Sea district, and as a token of appreciation for his continuous support, encouragement, and invaluable assistance provided to all of us throughout this study.

Holotype

NWHCM 1996.1, a complete pair of mandibles of an adult female individual, collected by Ian James Cruickshanks and Allister Cruickshanks on February 15th 1993, from the sandstone layer cropped out on the South Cliff at Easton Bavents, Suffolk County, East Anglia, England, UK (Fig. 1). The plastoholotype is deposited as NMR 7472 and NMNS-PV 23873.

Figure 1 The geologic context of Ontocetus posti sp. nov.

(A) Map of the locality of holotype mandible NWHCM 1996.1, Easton Bavents, East Anglia in southeast of the United Kingdom (based on data from Wood et al., 2009). (B) Stratigraphic column of the East Anglian Crags containing fossil horizon with relevant chronology (based on data from Daley & Balson, 1999; Wood et al., 2009). Maps and stratigraphic column made by M. Boisville and N. Chatar. Silhouette drawn by N. Chatar with Inkscape.

Formation and age

The Easton Bavents cliff comprises marine sands containing fossil shells and mammal bones, discovered at beach level or just beneath it (Holt-Wilson, 2015). The sandstone layer of the South Cliff site that produced the holotype corresponds to the base of the Norwich Crag Formation (Holt-Wilson, 2015) (Fig. 1). An age of 2.2–1.7 Ma is assigned to the Norwich Crag Formation, corresponding to the Early Pleistocene (Funnell & West, 1962; Daley & Balson, 1999; Westerhoff, 2009; Wood et al., 2009; Holt-Wilson, 2015; Boessenecker, Boessenecker & Geisler, 2018). This age is concurrent with the Dutch Tiglian Stage within the Norwich Crag Formation (Westerhoff, 2009; Holt-Wilson, 2015; King, 2016), representing a temperate climatic period known as the Antian-Bramertonian, approximately two million years ago. The sandstone layer where NWHCM 1996.1 was discovered has also yielded remains of fossil mammals washed out to sea, including falconer’s deer Eucladoceros falconeri, giant beaver Trogontherium sp., robust horse Equus robustus and proboscideans (Mammuthus meridionalis and Anancus arvernensis). Some other marine mammals like cetaceans (killer whale Orca and dolphins) have also been discovered (Holt-Wilson, 2015).

Referred specimen

IRSNB M156, partial left mandible from an adult male individual, featuring a fused anteromedial wall of the mandibular symphysis (Fig. 2), collected from Antwerp, Belgium approximatively during 1860s or 1870s by an unknown collector (van Beneden, 1876, 1877; Deméré, 1994b; Kohno & Ray, 2008; Boessenecker, Boessenecker & Geisler, 2018). Benthic foramina indicate correlation with zone NS44 and diatoms indicate correlation with zone DP2, providing an age of 3.71–2.5 Ma Late (Late Pliocene) (de Meuter & Laga, 1976; de Schepper, Head & Louwye, 2009; King, 2016). The exact locality of the specimen remains unclear, as van Beneden (1876) only mentioned various locations (Deurne district, Seefhoek district but also Fort I from Wyneghem district) all belonging to Antwerp Basin, Belgium. IRSNB M156 was initially described as a referred specimen of Trichecodon koninckii by van Beneden (1877), it was subsequently assigned to Odobenus koninckii (Deméré, 1994b) and eventually included in the hypodigm of Ontocetus emmonsi (Kohno & Ray, 2008).

Figure 2 Referred specimens assigned to Ontocetus posti.

IRSNB M156, the right mandible of a referred adult male (on the left side; from top to bottom: lateral view, occlusal view, medial view), and NMR 10087 (cast of RGM.St.119589), the symphysis portion of a referred subadult female (on the right side; from top to bottom: lateral view (mirrored), occlusal view, anterior view). Scale bar equals 5 cm. Pictures taken and figures drawn by M. Boisville.

RGM.St.119589, an incomplete fused mandible representing only the symphyseal portion, is assigned to a subadult female individual (Fig. 2). RGM.St.119589 was collected at the bridgehead near Domburg, Walcheren, Zeeland (The Netherlands) by an unknown member of the museum staff of the Rjiksmuseum van Geologie en Mineralogie from Leiden on June 26th 1961. Marine mammals collected from the seafloor, rivers, and estuaries in the vicinity of the mouth of the river Scheldt are likely derived from the Westkapelle/Brielle Ground Formation (Post & Bosselaers, 2005; Boessenecker, Boessenecker & Geisler, 2018). The Westkapelle/Brielle Ground Formation is Reuverian, corresponding to an age of 3.4–2.1 Ma (King, 2016; Boessenecker, Boessenecker & Geisler, 2018). The specimen was assigned to Odobenus huxleyi (Hooijer, 1957) with Lankester’s mention, corresponding to Trichecodon huxleyi (1865).

Remarks

The taxonomic history of Trichecodon koninckii is characterized by differing interpretations. Initially proposed by van Beneden (1871) based on a fragment of the left upper canine (original lost, but plastoholotype IRSNB 2892 survives) collected from the “Scaldisian sands” of Antwerp, Belgium, van Beneden (1877) later added a fragmentary mandible IRSNB M156 to this species as mentioned above. However, the original holotype, being a fragmentary tusk, lacks diagnostic character(s) for species-level identification. Rutten (1907) and van der Feen (1968) therefore dismissed T. koninckii as a nomen nudum due to this limitation. In contrast, Deméré (1994b) retained this species under the genus Odobenus based on the morphological characters of IRSNB M156 such as the fused symphysis, thin septa between postcanine tooth alveoli and diameter of lower canine greater than those of postcanine teeth. Kohno & Ray (2008) took a different approach, not following Deméré (1994a); instead, they included this referred mandible in the hypodigm of Ontocetus emmonsi, designating “koninckii” as a nomen dubium solely for the holotype tusk. In this study, we refrain from adopting either taxonomic stance for Trichecodon koninckii and IRSNB M156. This decision is motivated by the need to avoid confusion in species-level taxonomy and acknowledge the morphological disparities and diversification evident in the Pliocene of North Sea. Trichecodon huxleyi, originally proposed by Lankester (1865), was based on several fragmentary tusks from the Upper Pliocene deposits (Red Crag) in Suffolk, England. Deméré (1994b) retained this species under the genus Odobenus with Odobenus huxleyi based on cementum and outer orthodentine layer thinner than those of Odobenus rosmarus. Kohno & Ray (2008) considered it as synonymy of Ontocetus emmonsi only for the tusks.

Description

Mandible—NWHCM 1996.1 is an almost complete pair of mandibles (296.8 mm anteroposterior length, 67.2 mm transverse width at c1; see Table S1) (Figs. 3 and 4). The mandibular symphysis is fused, vascularized, particularly thicker anterodorsally and slightly thinner posteroventrally (49.0 mm symphysis length, 86.1 mm symphysis height) with a ratio reaching 1.75. In medial view, the mandibular symphysis is oval in shape and occupies approximately 29% of the maximum length of the mandible. In occlusal view, the lateral edge of the mandibular terminus and the symphysis form an angle of about 30° (see angle (e) in Table S1). The posteroventral margin of the symphysis reaches p2 in lateral view (Fig. 3). The anterior tip of the symphyseal region is not preserved but the shape of the anterior margin of the mandible suggests a slight convexity. The anterior margin must have been high, if we refer to the remnants at the base of the conical crown of the lower canine. In lateral view, the genial tuberosity is less developed anteroposteriorly, forming a small and smooth tubercle (Fig. 3). The anterior portion of the genial tuberosity reaches the anterior margin of the canine. A clear concavity is present between the posteriormost tip of the genial tuberosity and the anteriormost tip of the broken mandibular terminus, with an angle between the anterior and ventral margin (see angle (b) in Table S1) reaching 117°, corresponding to an extremely vertical condition of the anterior margin of the symphysis (Fig. 3). The ventral margin of the horizontal ramus is straight and starts at the level of p2. The dorsal margin of the horizontal ramus is concave and the angle between the anterior and dorsal margin (see Angle (a) in Table S1) reaches 131°. Mental foramina are present with a large oval medial mental foramen located behind canine, oriented antero-laterally and deeply hollowed out (Fig. 3). A second, smaller one, considered as an accessory mental foramen, is located posterior to the larger one and placed more ventrally. Its shape is circular, well hollowed and located beneath p2. Another circular-shaped accessory mental foramen is located anterior to p3, close to the occlusal outline, slightly hollowed out and opened posterolaterally. The anterior mental foramen is not visible in NWHCM 1996.1 due to the incomplete preservation of the anterior part of the mandible. In the occlusal view, the horizontal ramus is posteriorly divergent and laterally inflated, not presenting a sigmoidal occlusal outline (Fig. 3). In anterior view, the anterior extensions of both left and right genial tuberosities project anteriorly to the anteroventral surface of the symphyseal region, and therefore, its sagittal portion is slightly indented and makes a lateral furrow (Fig. 4). In the medial region of the horizontal ramus, a slightly swollen space can be observed, corresponding to the insertion of the mylohyoideus muscle (Fig. 4). Mylohyoideus insertion is delimited ventrally in NWHCM 1996.1. The mylohyoid line corresponds to the ventral limit of the mylohyoideus muscle, following an anteroposterior axis along the horizontal ramus and reaches p4. The mylohyoid groove corresponds to the posterior continuity of the mylohyoid line and is located next to the mandibular foramen. The superior mental spines are present, notably due to the fusion of the mandible (Fig. 4). These spines delimit the insertion area for the genioglossus muscle, one of the muscles of the tongue (Fig. 4). The inferior mental spines are also present and delimit the insertion area for the geniohyoideus muscle, one of suprahyoid muscles, which also includes the mylohyoideus muscle (Fig. 4). The ventral margin of the mandible corresponds to the insertion of another suprahyoid muscle, the digastricus muscle (Figs. 3 and 4). In lateral view, the digastric prominence is distinct, even if underdeveloped (Fig. 3). Ventral ridges seem to be present and represent the ventral part of the digastricus insertion zone (Fig. 3).

Figure 3 Holotype mandibles of Ontocetus posti in lateral view (left) and occlusal view (right).

NWHCM 1996.1, adult female (left), NMR 7472, cast of NWHCM 1996.1 (right). Scale bar equals 5 cm. The hatched parts correspond to broken surfaces. Pictures taken and figures drawn by M. Boisville.

Figure 4 Holotype mandibles of Ontocetus posti in anterior view (left) and posterior view (right).

NWHCM 1996.1, adult female (left), NMR 7472, cast of NWHCM 1996.1 (right). Scale bar equals 5 cm. The hatched parts correspond to broken surfaces. Pictures taken and figures drawn by M. Boisville.

The distance between the most posterior point of the dorsal margin and the base of the coronoid process in proportion to the entire mandibular length is short (~6%) (Fig. 3). A large part of the ascending ramus is missing and does not allow to establish the exact shape of the coronoid process. The masseteric fossa is well marked, large and located on the lateral side of the coronoid process. The fossa corresponds to the insertion area for the masseter pars profundus muscle (Fig. 3). The dorsal limit of the fossa is a ridge following the anteroposterior axis. This longitudinal surface above the ridge is tight. The superior posterior-most tip corresponds to the insertion area for the temporalis pars lateralis muscle, located at the apex of the coronoid process (Fig. 3). On the medial surface of the coronoid process, the pterygoid fossa is also large and well deepened. The pterygoid fossa corresponds to the insertion area for the temporalis pars medialis muscle (Fig. 4). A marked line in the anterior part of the fossa delimits the anterior border of the muscle. Despite the incompleteness of the ascending ramus, a vertical line is present, corresponding to the posterior limit of the temporal pars medialis muscle (Fig. 4). The ventral border of the pterygoid fossa is delimited by the mandibular foramen, an opening of the masseteric canal, connecting the mental foramina and the mandibular foramen. The mandibular foramen is circular in shape and opens posteriorly toward the mandibular condyle, with a cross-sectional area (Figs. 3 and 4). Ventrally to the mandibular foramen, a crest delimits the dorsal part of the pterygoideus medialis muscle (Fig. 4). The pre-angular space separates the ventral margin from the angular process through the digastric prominence. In NWHCM 1996.1, this space is short and straight. The angular process is located posterior to this space and is weakly developed. The articular facet of the angular process is projected posteroventrally and located ventrally to the mandibular condyle, close to the mandibular foramen. In medial view, the pterygoideus medialis muscle is inserted on the major part of the angular process. In lateral view, the condyloid crest is slightly prominent and corresponds to the insertion area for the masseter pars superficialis muscle ventrally, delimiting dorsally the masseter pars profundus muscle (Fig. 3). The mandibular condyle lies posteriorly to the condyloid crest and its most lateral portion is broken; however; what remains of its articular surface appears flat and not well developed (Figs. 3 and 4). The angle between the medial edge of the mandibular condyle and the mandibular symphysis in occlusal view (Fig. 3) (see Angle (d) in Table S1) is nearly perpendicular (95°). The mandibular notch, separating the coronoid process from the mandibular condyle, is long, giving an angle between the coronoid process and the mandibular condyle that is virtually vertical (see Angle (g) in Table S1). In occlusal view, the coronoid process is aligned with the cheek toothrow.

Dentition—The functional dental formula proposed for NWHCM 1996.1 is i2–3, c, p1–4. The specimen presents two incisors (i2 and i3). The root of the right i2 is preserved and located anteromedial to the canine with a narrow diastema (3.12 mm). A cavity just anterior to the canine is the alveolus for i3 (Figs. 3 and 4). It is sub-triangular and twice the size of the alveolus for i2. The canine is premolariform, followed by four cheek teeth. The cheek toothrow is almost straight but slightly laterally convex in occlusal view (Fig. 3), however the canine is not exactly aligned with the cheek teeth, located more medially especially than p3-4. The alveolar septum between c and p1 is reduced proportionally and septa between the cheek tooth alveoli are thin (4.3 mm). The canine is much larger in size than premolars, p1–p4 having equal-sized alveoli. The alveolus for i2 has a circular shape, while c is more oval in shape and p1–p4 nearly circular. All teeth are single-rooted, acuspid, and show no sign of enamel. The i2 is posteriorly projected and has an anteromedial wear facet. On the canine, the wear facet is anteromedial and inclined laterally while the wear on cheek teeth is anterolaterally at p1-2 and posteriorly at p3-p4. The right p1 is preserved with a lateral wear on its crown.

Comparisons

The sequence of changes in mandibular and dental morphology in odobenids is fairly well documented (Boessenecker & Churchill, 2013), the family being characterized by dental reduction and mandibular fusion associated with suction-feeding specializations (Figs. 5 and 6; see Fig. S1). Amongst the Odobenidae, species of the Odobeninae are characterized by having permanent postcanine tooth crowns showing no sign of enamel (at least it gets worn away very early in life), a vascularized mandibular terminus, and a divergent mandibular arch (Deméré, 1994b). Within odobenines, Aivukus cedrosensis can be distinguished by an elongated rostrum without an arched palate, C1 not enlarged as a tusk, presence of M1 and the lack of a central column of globular dentine. A monophyletic clade, the Odobenini, within the Odobeninae encompasses Protodobenus, Ontocetus, Odobenus and Valenictus and is characterized by a palate that is arched transversely and longitudinally; tusk-like upper canines with globular osteodentine core and premolariform C1 reduced in size. Protodobenus japonicus differs from other Odobenini by having two lower incisors transversely positioned, lower canine still caniniform and slightly reduced, a smaller upper canine and a shorter skull. Ontocetus is characterized by a combination of both primitive and derived characters such as an enlarged mastoid process, an elongated and upturned horizontal ramus, an unfused mandible, postcanine teeth widely spaced, and curved tusk-like C1, and it is separated from the monophyletic Odobenus+Valenictus clade. Odobenus and Valenictus are characterized by the fusion of the mandible, loss of lower incisors, genial tuberosity developed as a small tubercle, and an elongated and vaulted palate. Between them, Valenictus can be distinguished from Odobenus by its loss of dentition except C1 with an edentulous mandibular terminus considered as “a pad”, and a pachyosteosclerosis of postcranial bones. Based on these comparisons, NWHCM 1996.1, IRSNB M156 and RGM.St.119589 are safely recognized as belonging in the genus Ontocetus or Odobenus.

Figure 5 Hypothesized sequence of cranial, mandibular, and dental character transformations during odobenine evolution toward the suction-feeding specialization.

Cranial, mandibular and dental characters from Deméré (1994b) and Boessenecker & Churchill (2013). Scale bar equals 10 cm. The restored mandible of Ontocetus posti sp. nov. based on NWHCM 1996.1. The shaded areas correspond to the missing parts of the specimens. Reconstructions redrawn from R.W. Boessenecker illustrations by N. Chatar.

Figure 6 Significant mandibular and dental differences between Odobenus rosmarus, Ontocetus posti, and Ontocetus emmonsi.

Ontocetus emmonsi (A and B), Ontocetus posti (C and D), and Odobenus rosmarus (E and F). Female individuals are positioned on the left, male individuals on the right. From top to bottom: lateral view, occlusal view. Ontocetus emmonsi is represented by USNM PAL 9343 (referred female) and IRSNB M168 (referred male), Ontocetus posti is represented by NWHCM 1996.1 (referred female) and IRSNB M156 (referred male), Odobenus rosmarus is represented by IRSNB 1150B (female) and IRSNB 1150D (male). Pictures taken and figures drawn by M. Boisville.

The relative size of NWHCM 1996.1 and IRSNB M156 is similar to IRSNB M168, an Ontocetus emmonsi adult male mandible according to Kohno & Ray (2008). NWHCM 1996.1 and IRSNB M156 also exhibit other characteristics of the genus Ontocetus such as a lower canine much larger and separated from the cheek toothrow, and the presence of the first premolar (p1) within four postcanine teeth (p1–p4). McLaughlin et al. (2022) indicate that extant Od. rosmarus occasionally has four postcanine teeth, however, Fay (1982) shows that Od. rosmarus has lost p1 and its postcanine dental formula can only be summed up with three postcanine teeth as p2–p4. Moreover, NWHCM 1996.1 exhibits other traits similar to Ontocetus emmonsi such as a broad and short edentulous mandibular terminus with incisors, elevated and slanted anterior margin, a high, thin, and posterolaterally projected coronoid process, and short distance between the dorsal margin and coronoid process. However, NWHCM 1996.1 and referred specimens differ from Ontocetus emmonsi in some aspects, and share common features with Odobenus (Od. rosmarus), such as a fusion of mandible with symphyseal furrow. Deméré (1994b) recognized the fusion of the mandibular symphysis as a synapomorphy of Odobenus and Valenictus differentiating them from other odobenins (i.e., tribe Odobenini). However, dusignathine odobenids such as Dusignathus seftoni also have a fused symphysis, and this character is thought to have evolved several times and might be linked to the feeding strategy within the Neodobenia (i.e., Dusignathinae and Odobeninae). Nevertheless, even in the extant walrus Od. rosmarus, the fusion of the symphyseal region seems to show some degree of ontogenetic and individual variation, as explained in Taylor et al. (2020). IRSNB M156, the referred mandible, has also had a broad and fused symphysis (van Beneden, 1877, fig. 5, 6, 7; Fig. 2), sharing strong similarities with NWHCM 1996.1 and another referred mandible RGM.St.119589. The mandibular symphysis ratio for NWHCM 1996.1 (1.75) is closer to male individuals of extant Od. rosmarus and differs from high ratio measured in individuals of Ontocetus emmonsi (Table S1). In cross-section, the mandibular symphysis is oval in shape, similar to Odobenus, while more rectangular and elongated anterodorsally and posteroventrally in Ontocetus. The ratio between the size of the symphysis in NWHCM 1996.1 and the maximum length of the mandible is close to the values observed in extant Odobenus. In other words, the length of the symphysis of NWCHM 1996.1 but also in IRSNB M156 in occlusal view is relatively short like that of Od. rosmarus and much shorter than that of Ontocetus emmonsi represented by IRSNB M168 (Figs. 3 and 6; see Fig. S1). NWCHM 1996.1 and IRSNB M156 also share with Od. rosmarus, a horizontal ramus with a well-curved occlusal outline. The space between the canine and the cheek teeth, but also between the cheek teeth alveoli themselves, are also thinner in proportion, than in Ontocetus emmonsi, close to Od. rosmarus measurements. Lastly, NWHCM 1996.1 presents an underdeveloped angular process, and a flat and less developed mandibular condyle, different to the morphology presented in Ontocetus emmonsi. Based on those comparisons, identification of NWHCM 1996.1, IRSNB M156, and RGM.St.119589 as a distinct species within the genus Ontocetus is warranted. Thus, we recognize those mandibles as representatives of an unknown species and accordingly propose Ontocetus posti as a new species.

Morphometry

Our cluster dendrogram (Fig. 7) shows that the two specimens of Ontocetus posti (NMR 7472 and IRSNB M156) are similar to the female specimen of Ontocetus emmonsi (NMR 1890) this whole group clustering with the specimens of Odobenus rosmarus while the male specimens of Ontocetus emmonsi (IRSNB M168 and USNM PAL 475482) fall outside that cluster. Odobenus rosmarus occupies the highest values on the first axis of our PCoA while Ontocetus emmonsi and Ontocetus posti occupy the lowest values (Fig. 7). Axis two differentiates the two species of Ontocetus with On. posti occupying higher values than On. emmonsi, the higher values of On. emmonsi on axis two being measured in the female specimen. This morphospace suggested that On. posti was closer in terms of proportions to On. emmonsi than to Od. rosmarus which was later confirmed by our PERMANOVAs that retrieved a significant difference between On. posti and Od. rosmarus (p-value = 0.011) but not between On. posti and On. emmonsi (p-value = 0.2).

Figure 7 Cluster dendrogram (A) and morphospace occupation for the PCoA (B), based on mandibular ratios measured on different specimens of Odobenus rosmarus, Ontocetus posti and Ontocetus emmonsi.

See Supplemental Files for the ratios included. Silhouettes drawn by N. Chatar with Inkscape.

Phylogeny

Our phylogenetic analyses retried 46 equally parsimonious trees on which we performed a strict consensus. The strict consensus is available in Fig. 8. It is important to note that the position of On. posti does not vary on any of those most parsimonious trees. Ontocetus posti falls at the base of the Odobenus-Valenictus-Pliopedia clade, in between On. emmonsi and Od. rosmarus. Note that there are significant differences between the topology from Boessenecker et al. (2024) and our tree, notably in the topology for the ‘imagotariine’ walruses. The nodal support in our tree is relatively low with quite a few nodes with a symmetric resampling value above 50. We performed our analyses under implied weighting while Boessenecker et al. (2024) used equal weighting. Implied weighting reduces the weight of each character proportionally to its degree of homoplasy and those differences in topology therefore highlight the presence of numerous homoplastic characters in the matrix and the need for a revision of the walrus phylogeny. Consequently, the topology should be interpreted carefully except for the Valenictus-Pliopedia-Odobenus-Ontocetus-Protodobenus-Aivukus clade that is better supported.

Figure 8 Time-calibrated strict consensus cladogram of odobenid relationships based on the strict consensus tree (n = 46; best tree score = 597) recovered under implied weighting (with K = 12).

The shaded areas correspond to the missing parts of the specimens. Reconstructions redrawn from R.W. Boessenecker illustrations by N. Chatar.

Discussion

Generic allocation

Our phylogenetic analysis presented an odobenin with mosaic characteristics, exhibiting traits assignable to Ontocetus such as (1) a lower canine markedly larger than the cheek teeth (2) and it is rather separated from the cheek toothrow; (3) the presence of the lower first premolar (p1) with four lower postcanines considered as p1–p4; (4) a broad and short edentulous mandibular terminus with two lower incisors (i2, i3); (5) an elevated and slanted anterior margin, and (6) posterodorsally projected coronoid process. It also displayed characteristics of Odobenus, such as (1) the fusion of the mandibular symphysis; (2) an oval-shaped mandibular symphysis that is shorter in proportion to the total length of the mandible; (3) a horizontal ramus with a well-curved lateral occlusal outline; and (4) a shorter space between the canine and the cheek teeth. Morphometric results suggested that this mosaic-character combination within the odobenins positioned significantly closer to Ontocetus than to Odobenus (Fig. 7). Phylogenetic analyses suggested that our specimens are sister to the Valenictus + Odobenus clade (Fig. 8). These results indicate that On. posti might be more closely related to the later diverging species including Valenictus and Odobenus, potentially indicating an ancestral morphology for the first representative of Valenictus + Odobenus clade, absent from the North Atlantic during the Early Pleistocene. Although our result did not directly support parallel evolution of On. posti with Odobenus, it is apparent from previously proposed odobenin synapomorphies that such derived characters of On. posti shared with Valenictus + Odobenus clade could be considered to be strongly related to their suction-feeding behavior. Thus, while the present analysis recognizes morphological synapomorphies for On. posti and Valenictus + Odobenus clades, potential homoplasies for such characters directly related to their feeding behavior within the clade limit their diagnostic value.

By contrast, the oldest records of the genus Odobenus correspond to an isolated tusk from the Pliocene-Pleistocene Shiranuka Formation of Japan (Kohno et al., 1995) and a nearly complete Late Pliocene skull from the bottom of the Sea of Okhotsk (Miyazaki, Kimura & Ishiguri, 1992). Mandibles assigned to Odobenus mandanoensis (ca. 0.5 Ma) (Tomida, 1989), Odobenus rosmarus (ca. 0.7 Ma) (Kimura et al., 1983; Ishiguri & Kimura, 1993), and a partial skull from the Yabu Formation (Chiba Prefecture) from the Middle Pleistocene strata (Miyazaki et al., 1995) have also been recorded from the North Pacific realm. Although some cervical vertebrae of Odobenus sp. have been mentioned by Brigham (1985) from the Gubik Formation (Alaska), it is unclear to estimate the origin of these remains, as the author seems to mention several sections for Odobenus remains, from Aminozone 2 (ca. 0.5 Ma) and/or Aminozone 3 (1.7–0.7 Ma), and does not provide any illustration of vertebrae. The oldest known specimen assigned to Odobenus rosmarus outside the North Pacific comes from Northern European Russia (Pechora River), dated to the end of the Middle Pleistocene (Ponomarev et al., 2023), potentially suggesting a dispersion outside the North Pacific at least during the latter half of the Middle Pleistocene. In contrast, Ontocetus was already present along the western coasts of the North Atlantic during the Early Pliocene (Berry & Gregory, 1906; Kohno & Ray, 2008), with occurrences in the North Sea in the early Late Pliocene (e.g., Lankester, 1865; du Bus, 1867; van Beneden, 1871; Hasse, 1909) until the Early Pleistocene (Erdbrink & van Bree, 1990). Remains assigned to Ontocetus have been found in the late Late Pliocene Moroccan deposits of Ahl al Oughlam (Geraads, 1997), and a final population may have survived on the east coast of the United States until about 1.1 Ma (Boessenecker, Boessenecker & Geisler, 2018). Considering these elements, and pending more field exploration of Pliocene–Middle Pleistocene fossil localities within the Arctic Circle, particularly the Gubik Formation (Alaska), but also more odobenin material from the Early Pleistocene of the North Sea, it is preferable to assign these specimens to Ontocetus, presenting an independent evolutionary convergence of suction-feeder with Odobenus.

Functional interpretations toward suction-feeding specialization

While the first odobenids were piscivorous, they rapidly evolved toward molluscivory (Adam & Berta, 2002), with some taxa, like Valenictus chulavistensis, exhibiting an extreme reduction of dentition except for the upper canines just as the extant walrus (Deméré, 1994a). Ontocetus, previously considered a molluscivore (Deméré, 1994a, 1994b; Adam & Berta, 2002; Boessenecker, Boessenecker & Geisler, 2018), however, exhibits features suggesting less specialization, making its diet a bit more complex to assess. Odobenus rosmarus exhibits a unique combination of characters that optimize its ability to engage in suction-feeding (Kastelein & Gerrits, 1990; Kastelein, Gerrits & Dubbeldam, 1991). NWHCM 1996.1 and IRSNB M156 exhibit characteristics specific to Ontocetus, such as a significantly larger lower canine compared to the cheek teeth, four post-canine teeth, and incisors (i2). However, some features of those specimens are also found in Odobenus including mandibular fusion and shortening, oval mandibular arch, less developed and flatter mandibular condyle, and thin septa between the various teeth. The robustness of the mandibular symphysis is directly associated with its condition, subjected to torsional forces experienced during feeding (Adam & Berta, 2002). Mandibular fusion, concomitant with symphyseal size reduction, serves as a preventive measure against substantial mandibular deformation, particularly during tongue retraction in suction-feeding scenarios (Gordon, 1984; Kastelein & Gerrits, 1990; Deméré, 1994a; Adam & Berta, 2002). The fusion of the mandible and the pachyosteosclerotic reinforcement of the anterior part of the mandible further contributes to enhanced resistance, by supporting the large tusks. Suction-feeding specialization is further evidenced by the dorsally concave vaulting of the hard palate in the Odobenini, evident in both transverse and longitudinal planes. This palatal configuration, as proposed by Kastelein & Gerrits (1990), results in an enlarged intraoral space, facilitating a more extensive tongue protraction before retraction during suction-feeding. The curvature and oval shape of the mandible in occlusal view correspond to the arched structure of the hard palate vaulting. Additionally, the shortening of the skull, and consequently of the mandible, enhances the containment of negative intraoral pressure, facilitating the suction of mollusks when water reaches the palate (Werth, 2006a, 2006b).

Concurrently, the reduction in dental series among the Odobenini, as documented by Boessenecker & Churchill (2013), manifests in Odobenus with narrower spacing between teeth compared to Ontocetus. This dental morphology is likely a consequence of Odobenus specialization in suction-feeding, accompanied by diminished reliance on teeth and a relatively free of obstruction feature into the oral cavity that would enhance the efficiency of suction-feeding (Adam & Berta, 2002). Additionally, the reduced and flattened mandibular condyle in Odobenus is indicative of an adaptation to suction-feeding, paralleling the morphological similarity observed in Valenictus, identified as the most specialized suction-feeder among the Odobenini (Deméré, 1994a). This suggests that Ontocetus posti was able to use suction to catch its prey and may have once occupied an ecological niche similar to the extant Odobenus rosmarus in the Pliocene of the eastern North Atlantic before the latter species came into the North Atlantic (Fig. 9).

Figure 9 Paleobiogeographic distribution of the subfamily Odobeninae and trend towards specialization to suction-feeding in both North Pacific and North Atlantic Ocean.

Bold line: temporal range of the species; arrows: independent migration from the North Pacific to the North Atlantic. Geochronological ranges after Kohno et al. (1995), Magallanes et al. (2018) and Boessenecker et al. (2024). The shaded areas correspond to the missing parts of the specimens. Reconstructions redrawn from R.W. Boessenecker illustrations by N. Chatar.

Suction-feeding is observed independently in various groups of marine mammals, starting with pinnipeds, as in the bearded seal Erignathus barbatus. Nevertheless, its adaptations differ somewhat. The rostrum of the bearded seal is flatter, highly mobile, and capable of diverse shape changes (Marshall, Kovacs & Lydersen, 2008; Marshall, 2016). The bearded seal retains incisors and possesses a relatively flat palate, although the maxillary alveolar processes are expanded, resulting in a concave ventral rostral surface in transverse section. Adam & Berta (2002) propose that Erignathus barbatus may employ suction-feeding, possibly to a lesser extent than Odobenus rosmarus, although this has yet to be confirmed with experimental data. Other pinnipeds, such as the leopard seal Hydrurga leptonyx and the harbor seal Phoca vitulina, employ suction for prey capture (Hocking, Evans & Fitzgerald, 2013; Marshall et al., 2014; Churchill & Clementz, 2016; Kienle & Berta, 2016; Hocking et al., 2017), albeit without undergoing modifications as significant as those observed in the mandible and skull of Odobenus rosmarus. Suction-feeding is also present independently in contemporary toothed whales and dolphins (beaked whales, monodontids and sperm whales), involving the loss of elongated mandibles and skulls, accompanied by a reduction in teeth (Heyning & Mead, 1996; Werth, 2000; Kane & Marshall, 2009). This specific feeding technique is also surprisingly present in baleen whales, in the sole gray whale Eschrichtius robustus, predominantly employing this capability for aspirating prey-laden sediment residing on the seabed. This behavior aligns more closely with suction-filter feeding than a complete suction-feeding specialization (Ray & Schevill, 1974).

In the fossil record, suction-feeding has been proposed with the cetacean Odobenocetops (de Muizon & Domning, 2002), possessing an arched and vaulted palate, distinctive of this feeding specialization. Furthermore, de Muizon & Domning (2002) suggested that tusks also facilitate suction-feeding, acting as a “sled” in order to maintain a useful distance between the gape and the benthic substrate, so that clams or other mollusks can be more efficiently sucked up and harvested from the seafloor. In Odobenus rosmarus, tusks have often been considered as a function of hauling out of water or digging in ice for holes (Kastelein & Gerrits, 1990), however it would appear that tusks were used mainly as secondary sexual characteristics, with diversity of shape and size throughout the evolutionary history of the Neodobenia, linked to mating behaviour (lekking and/or female-defense polygyny) and used as a display and in rarer cases combat (Deméré, 1994a; Boessenecker & Churchill, 2021). Although Fay (1982) demonstrated that extant walruses did not use their tusks for benthic feeding, recent observations of males linked abrasive wear on the anterior surface of tusks with the tusks being dragged along the sediment during feeding in a sledge-like manner (Levermann et al., 2003), while anterior surface spanning the length of the tusks was not observed in any of the known female specimens. This observation corroborates sexually dimorphic feeding behaviors (McLaughlin et al., 2022). According to Born et al. (2003), male individuals predate larger bivalves than females. It may possible that behavioral and ecological sexual dimorphism exist in Od. rosmarus, with females potentially foraging in shallower water, outside same beaching areas than males (Lydersen, 2018). It is quite difficult to estimate the exact function of the tusks in Ontocetus, given that they are shorter and more procumbent than those of Od. rosmarus. In the absence of clear evidence, it is preferable to consider the primary function of the tusks of Ontocetus as secondary sexual characteristics, mainly used by the males for display or combat during female-defense polygyny. However, further studies are necessary to assess whether Ontocetus would have had sexually dimorphic feeding behavior like Od. rosmarus (McLaughlin et al., 2022), especially in view of the potential geographical sexual segregation that would have existed in Ontocetus (Kohno & Ray, 2008). NWHCM 1996.1 demonstrates distinctive muscular insertions, particularly evident on both the horizontal and vertical ramus. Observable variations are discernible among Odobenus rosmarus, Ontocetus posti, and Ontocetus emmonsi (Fig. 10). On the horizontal ramus, the hyoid muscle insertion zones can be identified, bounded ventrally by the mylohyoid line (anteriorly) and the mylohyoid groove (posteriorly), proximate to the mandibular foramen. To some extent, the insertion zone for the mylohyoideus muscle appears broader in Ontocetus compared to Odobenus rosmarus. Specimen preservation enables the observation of inferior mental spines, corresponding to the geniohyoideus muscle insertion, and, to some extent, superior mental spines, corresponding to the genioglossus muscle insertion. Ontocetus appears to possess proportionally more developed genioglossus and geniohyoideus muscles. The digastricus muscle can be estimated, characterized by the presence of a digastric prominence. However, Ontocetus posti exhibits a less pronounced digastric prominence, resembling that of Odobenus rosmarus. On the vertical ramus, the masseteric fossa is notably more pronounced in Ontocetus compared to Odobenus rosmarus. This fossa corresponds to the muscular insertion area of the masseter profundus muscle. Ventral to the condyloid crest, the muscular insertion area of the masseter superficialis appears distinguishable, with Ontocetus emmonsi having a proportionally larger muscular insertion area. Consequently, it is plausible that Ontocetus emmonsi possesses a more developed masseteric muscle compared to Ontocetus posti and Odobenus rosmarus. The pterygoid fossa on the medial face of the coronoid process also appears more pronounced in Ontocetus compared to Odobenus rosmarus. This fossa corresponds to the insertion zone of the temporalis muscle, differentiating into the temporalis pars medialis in the medial zone of the coronoid process and the temporalis pars lateralis located at the apex of the process. Ontocetus also exhibits a proportionally larger zone and, consequently, a more developed temporal muscle than Odobenus rosmarus. Lastly, the insertion zone of the pterygoideus muscle, specifically its medial part (pterygoideus medialis), can be observed. This zone appears slightly smaller in size in Ontocetus emmonsi compared to Odobenus rosmarus and Ontocetus posti. In general, the comparison and interpretation of muscle functions between the fossil record and the present pose challenges. Considering the proportions of muscular insertion zones found on the mandibles of Ontocetus and Odobenus rosmarus, it appears that fossil taxa have slightly larger muscular insertion areas in proportion, potentially indicative of more developed muscles. This suggests a possible moderately lesser specialization toward suction-feeding, given the different proportional changes in muscular insertion zones related to this mechanism.

Figure 10 Anatomical interpretations for muscle insertions in Odobenus rosmarus (A, B), Ontocetus posti (C, D), and Ontocetus emmonsi (E, F).

Lateral view on the left side, posterior view on the right side. Scale bar equals to 5 cm. IRSNB 1150D (male morphotype) represents Odobenus rosmarus, NWHCM 1996.1 (female morphotype) represents Ontocetus posti, IRSNB M168 (male morphotype) represents Ontocetus emmonsi. Pictures taken and figures drawn by M. Boisville.

Although we have attempted to describe these insertion zones, Kienle, Cuthbertson & Reidenberg (2021) conducted a comparative examination of craniofacial musculature in pinnipeds and its role in aquatic feeding. Contrary to their initial hypothesis, approximately half of the biting species in this study (i.e., harbor seals and ringed seals) do not exhibit specific musculoskeletal adaptations for biting or suction-feeding. Moreover, several pinniped species without morphological adaptations for suction have been shown to be extremely capable suction-feeders (e.g., Hocking, Evans & Fitzgerald, 2013; Marshall et al., 2014; Kienle et al., 2018). Thus, it seems that most pinnipeds are opportunistic predators, and their non-specialized craniofacial musculature allows flexible feeding behavior. In light of this information, it is more likely that osteological changes (mandible fusion, reduction of teeth, skull shortening, or vaulted and arched palate) are specific to the walrus lineage in the specialization for suction-feeding, rather than representing a change in craniofacial musculature.

In this regard, Ontocetus emmonsi is inferred to be moderately less specialized in suction-feeding based on observations of a lower canine wider than the cheek teeth, the presence of incisors and the absence of mandibular fusion. On the other hand, some features observed in both Ontocetus posti and the extant walrus such as the presence of a fused and reduced mandibular symphysis and a well-curved mandibular arch, attesting of a shorter skull, may imply that this taxon might have been more specialized in suction-feeding than previously thought for the genus Ontocetus. Indeed, the absence of Odobenus rosmarus left this specialized molluscivore/suction-feeder niche open for Ontocetus in the Early Pleistocene of the North Sea. We hereby suggest that Ontocetus posti might have occupied a similar niche to Odobenus rosmarus. It is still interesting to note that despite those adaptations to suction-feeding more marked than in Ontocetus emmonsi, the mandibular and dental proportions of Ontoceus posti are still much more similar to Ontocetus emmonsi than Odobenus rosmarus (Fig. 7).

Sexual dimorphism in Ontocetus posti

Sexual dimorphism is a key morphological characteristic in pinnipeds, usually associated with their polygynous behavior (Bartholomew, 1970; Kovacs & Lavigne, 1992; Garlich-Miller & Stewart, 1998; Weckerly, 1998; Lindenfors, Tullberg & Biuw, 2002; Ralls & Mesnick, 2009; Jones & Goswami, 2010; Velez-Juarbe, 2017; Mesnick & Ralls, 2018). In Odobenus rosmarus, adult males exhibit conspicuous differences in size and weight compared to adult females, and several studies highlight sex-dependent morphological variations in their crania and mandibles (Mohr, 1942; Fay, 1982; Kovacs & Lavigne, 1992; Garlich-Miller & Stewart, 1998; Boisville et al., 2022). Among the odobenins, Ontocetus emmonsi has been hypothesized to display extreme sexual dimorphism, even more marked than in Od. rosmarus (Kohno & Ray, 2008). Larger individuals (males) are approximately 20–40 percent larger than smaller individuals (females) according to cranial, mandibular and postcranial sizes, based on the sample of the Yorktown Formation (Lee Creek Mine, North Carolina, USA; Kohno & Ray, 2008). Similarly, a clear difference in upper canines indicates distinctions between males (larger and stocky tusks) and females (smaller and slender tusks).

Consequently, it is justified to investigate the sexual dimorphism in Ontocetus posti. The size of mandible as well as the circumference of the lower canines in NWHCM 1996.1 corresponds to the dimensions of male specimens of Ontocetus emmonsi (IRSNB M168, ROM 26116, USNM PAL 475482). Although incomplete, IRSNB M156 also exhibits proportions consistent with the male morphotype of Ontocetus emmonsi, whereas RGM.St.119589 displays proportions much similar to the female morphotype (USNM PAL 9343). It is noteworthy that IRSNB M156 presents measurements larger than any documented Ontocetus. Males and females can be easily distinguished using the symphysis height/symphysis length ratio, with females exhibiting a lower ratio indicative of a slender mandible (Kohno & Ray, 2008). IRSNB M156 exhibits values similar to the male Ontocetus emmonsi, while NWHCM 1996.1 and RGM.St.119589 exhibit values closer to females. Mandibular height (see MD in Table S1) and mandibular width (see MT in Table S1) after the last cheek tooth, indicate that NWHCM1996.1 aligns with male Ontocetus emmonsi values, and IRSNB M156 surpasses these values, suggesting an increased mandibular thickness. The length of the cheek toothrow and the length between the last cheek tooth and ascending ramus (see Table S1) is also sexually dimorphic. NWHCM 1996.1 appears closer to female Ontocetus emmonsi values (USNM PAL 9343), whereas IRSNB M156 exhibits broader proportions, particularly with a longer cheek toothrow than the largest males. Furthermore, transverse width of the mandible at the lower canine is narrower in NWHCM 1996.1 and RGM.St.119589, closer to female Ontocetus emmonsi values (USNM PAL 9343, USNM PAL 374273), while IRSNB M156 reaches male Ontocetus emmonsi values. A narrow width between the lower canines may reflect a narrower space between the upper canines and thus a relatively narrow rostrum, corresponding to a female morphotype in walruses (e.g., Boisville et al., 2022; Boessenecker et al., 2024). Regarding ontogenetic age, NWHCM 1996.1 and IRSNB M156 should be regarded as adult individuals, based on their overall size, complete tooth eruption, tooth wear facets and complete fusion of the symphysis, while RGM.St.119589 should be considered as subadult based on the smaller relative dimensions of its teeth (i.e., thinner cementum of tooth roots) and the presence of a distinctive symphyseal furrow still present on the mandible, indicating that the fusion was not entirely complete.

Based on these comparisons, we identify NWHCM 1996.1 as an adult female, RGM.St.119589 as a subadult female, and due to its extreme proportions, IRSNB M156 is recognized as an old adult male, likely considered the largest Ontocetus individual ever found. Ontocetus posti may have had a significant sexual dimorphism, like Ontocetus emmonsi. However, intraspecific variation with respect to the mandibles of Ontocetus posti appears to be less than Ontocetus emmonsi, approaching that found in Odobenus rosmarus.

Biochronological dispersal and evolution of Ontocetus in the North Atlantic

The youngest record of Ontocetus in the North Atlantic is dated to the Early Pleistocene (1.8–1.1 Ma) in the Upper Waccamaw Formation deposits (Austin Sand Pit, South Carolina, USA; Boessenecker, Boessenecker & Geisler, 2018). Although the Easton Bavents specimen is slightly older (2.2–1.7 Ma), it represents the final occurrence of this genus in the North Sea. The biochronology and dispersal of Ontocetus is well-established, with its initial appearance traced back to the Upper Miocene/Lower Pliocene boundary age Tatsunokuchi Formation (Okamoto & Kohno, 2019) and Lower Pliocene Joshita Formation in Nagano Prefecture, Japan (Kohno, Narita & Koike, 1998). Additional specimens have been uncovered in contemporaneous deposits in Japan (Yasuno, 1988; Kohno et al., 1995). This suggests that Ontocetus originated in the North Pacific; those more primitive forms also seem distinct from those discovered in Lower Pliocene deposits on the United States east coast or in the North Sea (Kohno et al., in prep.).

The dispersal of Ontocetus from the North Pacific to the North Atlantic has traditionally been proposed to have occurred via the Central American Seaway around the Mio-Pliocene boundary (Repenning, Ray & Grigorescu, 1979; Kohno et al., 1995; Kohno & Ray, 2008). Another dispersal route via the Arctic through Bering Strait has also been hypothesized (Deméré, Berta & Adam, 2003; Boessenecker, Boessenecker & Geisler, 2018; Boessenecker & Churchill, 2021). This perspective suggests that Ontocetus evolved during the Late Miocene in the North Pacific before migrating to the North Atlantic. The earliest occurrence of Ontocetus in the North Atlantic comes from the Upper Bone Valley Formation in Florida (e.g., Kohno & Ray, 2008; Boessenecker, Boessenecker & Geisler, 2018). Subsequent deposits presenting Ontocetus appear in a geochronostratigraphic succession (Yorktown Formation in North Carolina and Virginia; Lower Tamiami Formation in Florida; Goose Creek Limestone and/or Raysor Formation in South Carolina; Pinecrest Beds of the Tamiami Formation in Florida) at similar or more northern latitudes, potentially indicative of a distribution that expanded from south to north (Kohno & Ray, 2008). Considering the corresponding latitudinal occurrences in Early Pliocene deposits (Florida, South Carolina, North Carolina), Ontocetus may appear to be adapted to warmer temperate water than the extant species, especially given the absence of Ontocetus occurrences at high latitudes, although the absence of fossiliferous rocks at these latitudes may represent a bias rather than a signal.

Additionally, an extinct baleen whale Herpetocetus has very often been found in Upper Miocene to Pliocene similar deposits associated with odobenins (i.e., Purisima Formation, California, USA; Tatsunokuchi Formation, Sendai, Japan; Yorktown Formation, North Carolina and Virginia, USA; Lillo Formation, Belgium; San Diego Formation, California, USA). However, Herpetocetus already appears to be present at least in Upper Miocene deposits of Belgium although the origin of the specimen remains undecided (El Adli, Deméré & Boessenecker, 2014). The potential presence of Herpetocetus as early as the Late Miocene in the North Sea also argues for a later arrival, probably via the southern western North Atlantic, rather than the Arctic, for Ontocetus.

Although there is a lack of fossils of Ontocetus from the entire Pacific coast of North America to date, which seems to have had its own faunal province with little migration events (Boessenecker, 2013), the presence of an another odobenin Valenictus (Late Miocene up to Late Pliocene) on the west coast of the USA (California) supports a more southerly migration of Pliocene tusked walruses via the Central American Seaway rather than through the Arctic (Repenning & Tedford, 1977; Deméré, 1994a, 1994b). The gregarious behavior, the dependence on shallow embayment along the coasts, their oldest occurrence in the western North Atlantic located in Florida, the absence of Pliocene and Lower Pleistocene fossiliferous deposits at high latitudes in the western North Atlantic and the fact that this does not significantly predate the closure of the Central American Seaway (Grossman et al., 2019), may suggest that Ontocetus may have migrated from the North Pacific to the North Atlantic via the still-open Central American Seaway at the time (Naafs et al., 2010; O’Dea et al., 2016) (Fig. 11A). More information on the understudied Pliocene-Middle Pleistocene fossil localities within the Arctic Circle (e.g., Gubik Formation, Alaska) could help clarify the migration pattern of Ontocetus (Brigham, 1985; Boessenecker, Boessenecker & Geisler, 2018).

Figure 11 Ontocetus biogeographic hypothesis, and reconstruction of Ontocetus posti occurrences during the Early Pleistocene in the North Sea (2.2–1.7 Ma).

(A) the Late Miocene-Early Pliocene map: (1) evolution of Ontocetus emmonsi from western North Pacific followed by dispersal through the Central American Seaway to the western North Atlantic during the Early Pliocene; (B) the late Early Pliocene map: (2) dispersal of Ontocetus emmonsi to the eastern North Atlantic during interglacial periods; (C) the Late Pliocene-Early Pleistocene map: (3) the dispersal to Morocco during glacial periods, and isolation of an Ontocetus population in the North Sea, or (4) alternatively southerly expansion or emigration from the supposed area of the paleo-Bay of Biscay to Morocco; (D) Early Pleistocene map: reconstruction of Ontocetus posti occurences in the North Sea (2.2–1.7 Ma). Ontocetus biogeographic hypothesis proposed by Repenning & Tedford (1977), Kohno et al. (1995), Geraads (1997) and Kohno & Ray (2008). Ice-sheets and sea shores reconstructed from Meijer & Preece (1995), Van Vliet-Lanoë et al. (2002), Naafs et al. (2010), Gibbard & Cohen (2015), Gibbard & Lewin (2016), Rea et al. (2018), Batchelor et al. (2019), Westerhoff et al. (2020) and Lein et al. (2022). Maps and silhouettes modified from Boessenecker & Churchill (2021). Pliocene occurrences are represented by orange color, Pleistocene occurrences are represented by blue color.

Ontocetus then dispersed along the western North Atlantic during the Early Pliocene, populating fossil-rich formations such as the Bone Valley Formation, Raysor Formation, and Yorktown Formation (Berry & Gregory, 1906; Kohno & Ray, 2008). However, around 4.6–4.2 Ma, the closure of the Panama Isthmus disrupted equatorial currents between the Pacific and Atlantic Oceans, leading to the cooling of the Atlantic Ocean and eventual ice cap formation (O’Dea et al., 2016). Between these glaciations, during the warm periods around the late Early Pliocene (3.8 Ma), a population from the western North Atlantic may have migrated towards the North Sea. This migration was facilitated by the intense North Atlantic current transporting warm waters northward and maintaining the higher latitudes’ warmth (Naafs et al., 2010) (Fig. 11B). However, during the Late Pliocene (3.1 Ma), sea-level oscillations, likely acting in conjunction with other oceanographic alterations such as changes in productivity, ocean circulation, and biotic drivers such as prey availability, impacted neritic areas and led to the extinction of the Pliocene marine megafauna (Pimiento et al., 2017). The loss of abundant shallow embayments and seaways, coupled with faunal turnover in the Late Pliocene, likely contributed to the disappearance of mid-latitude walruses, which were highly specialized in feeding adaptations (Boessenecker, 2013, Boessenecker & Churchill, 2013; Pimiento et al., 2017).

No occurrences of otariids or phocids were found in the North Sea during the Pliocene, except a monachinae indet. (Dewaele, Lambert & Louwye, 2018b), and two undescribed occurrences (cf. Pagophilus sp., cf. Pusa sp.; Post & Bosselaers, 2005). Given the robust morphology of its forelimb, giving highly moving abilities by robust limbs for taking up breeding territories by example, as well as its craniomandibular anatomy (relative elongation of the skull, characterized by a more slanted cranial roof, non-fused and longer mandible, more complete dentition, particularly with functional incisors, and a more developed mandibular condyle) Ontocetus emmonsi has been proposed to have occupied a close ecological niche of the current sea lion who was absent from the North Pacific until the end of the Pliocene (Ontocetus sp., Okamoto & Kohno, 2019).

The global cooling during the Plio-Pleistocene transition had a profound impact on the warm North Atlantic Current, causing the ice cap to extend to lower latitudes (Naafs et al., 2010; Batchelor et al., 2019). During the Late Pliocene, a North Sea population of Ontocetus may have evolved and specialized in suction-feeding techniques, particularly in response to changes in mollusk fauna dominated by immigrants from the Pacific Ocean (Macoma balthica) and other bivalves (Riches, 2010). In the cold periods of the Early Pleistocene, the warm North Atlantic Current did not flow northwards but rather west-eastwards, directly towards Morocco, where the Arctic ice cap likely reached Spain (Naafs et al., 2010). Ontocetus has also been identified in the Ahl al Oughlam deposits of Morocco through the Plio-Pleistocene boundary (3.0–2.2 Ma) (Alachtherium africanum Geraads, 1997) (Fig. 11C). This raises the possibility that the Moroccan population of Ontocetus became endemic due to the altered direction of the North Atlantic Current and ice cap expansion. A population of Ontocetus from the western North Atlantic may have migrated toward Morocco, or that part of the eastern North Atlantic population may have migrated southwards through the paleo-Bay of Biscay toward Morocco, although there is no fossil evidence at present (Fig. 11C). Later, as the Early Pleistocene unfolded, warm periods became increasingly rare, contributing to an overall colder climate with drastic falls in sea levels due to ice cap expansion.

Extinction of Ontocetus in relation to global cooling through Early Pleistocene

Despite its suction-feeding adaptation to the exclusive molluscivorous diet in the North Sea, Ontocetus posti became extinct around 1.7 Ma. This extinction may also be explained by the confinement of the taxa in the North Sea and unable to escape northwards to the Atlantic Ocean due to ice sheet expansion (Van Vliet-Lanoë et al., 2002; Gibbard & Cohen, 2015; Gibbard & Lewin, 2016; Rea et al., 2018; Lein et al., 2022) (Fig. 11D). Moreover, the Dover Strait was closed and would not have reopened until the middle of the Chibanian or Holsteinian of about 400 Ka (Meijer & Preece, 1995; Westerhoff et al., 2020). Ontocetus posti may have been overly specialized in suction-feeding for molluscivorous diet, rendering it more vulnerable to rapid changes in sea level oscillation and the general global cooling during this period. Additionally, during the Late Tiglian (1.8 Ma), there was a significant drop in mollusk species, with few "warm" mollusks (Meijer & Preece, 1995). The Late Pliocene marine mammal assemblage in the southern North Sea also includes other species coexisting with Ontocetus, some of which using a similar suction-feeding technique, such as piscivorous Mesoplodon sp. (Ziphiidae), Delphinapterus sp. (Monodontidae), and Globicephala sp. (Delphinidae) (de Vos, Mol & Reumer, 1998; Post & Bosselaers, 2005). The only other pinnipeds found are two Phocidae (cf. Pagophilus sp. and cf. Pusa sp.) and a monachinae indet., which, based on their contemporary representatives, did not occupy the same niche as Ontocetus (Post & Bosselaers, 2005; Dewaele, Lambert & Louwye, 2018b).

Interestingly, one final population of Ontocetus emmonsi managed to survive at much lower and warmer latitudes in the Upper Waccamaw Formation, South Carolina, USA, until 1.1 Ma (Boessenecker, Boessenecker & Geisler, 2018). Although this is at present evidenced only by the fossil record of a relatively short and strongly curved tusk, it may suggest that the oral cavity or the rostrum of this animal had still retained a relatively generalized one. As a result, it might have been facilitated much broader dietary preferences by a less specialized feeding apparatus and suffered less quickly or significantly suffered the environmental changes during the Pliocene and Early Pleistocene. More stable and favorable climatic conditions, at more southerly latitudes compared to the North Sea, also enabled Ontocetus to survive until 1.1 Ma.

Further findings on this final population are still needed to better recognize its species-level assignment. The temporal separation of Odobenus and Ontocetus in the North Atlantic for approximately one million years suggests that they did not compete for resources and reflects separate Early Pliocene (Ontocetus) and Middle–Late Pleistocene (Odobenus) dispersals into the North Atlantic from the North Pacific. These dispersals occurred most likely through the Central American Seaway for the warmer-tolerant Ontocetus and the Arctic for the colder-tolerant Odobenus (Kohno et al., 1995; Kohno & Ray, 2008).

Conclusions

We described a novel taxon of Ontocetus, identified through a re-evaluation of a nearly complete pair of mandibles from the Lower Pleistocene of Easton Bavents (United Kingdom) and a fragmentary mandible from the Upper Pliocene of Belgium. Key diagnostic features, such as the presence of four post-canine teeth including p1, a lower canine larger than the cheek-teeth, and an i2 clearly characterize the extinct genus Ontocetus. However, Ontocetus posti shares features with the extant Odobenus, such as a fused and shorter symphysis, a well-curved horizontal ramus and thin septa between the canine and cheek-tooth alveoli. Some of those characters being associated with suction-feeding adaptation in the extant walrus, we hereby suggest that Ontocetus posti occupied an ecological niche similar to the extant Odobenus rosmarus in the Late Pliocene of the eastern North Atlantic.

Ontocetus and Odobenus did not coexist in the North Atlantic, Odobenus appearing almost a million years later subsequent to the extinction of Ontocetus. Described as a temperate to warm-tolerant tusked walrus, Ontocetus colonized the western North Atlantic during the Early Pliocene before migrating to the North Sea during the warm periods of the Late Pliocene. Its prevalence in warmer waters contrasts with the genus Odobenus. Global climatic cooling during the Early Pleistocene invariably impacted North Sea mollusk faunas and contributed to the isolation of the North Sea from the North Atlantic. Those extrinsic factors associated with the specialization in suction-feeding of Ontocetus posti likely contributed to its extinction around 1.7 Ma. A more in-depth investigation holds promise in unveiling the past diversity of Ontocetus and Odobenus on a global scale, elucidating the eventual dominance of the cold-tolerant Odobenus rosmarus as the sole survivor.

Supplemental Information

Supplemental Information 1 R Codes regarding New Odobenids remains phylogeny and morphometry.

Supplemental Information 2 Ontocetus cladistic matrix from this study (Nexus).

The raw results of the phylogenetic analyses.

Supplemental Information 3 Ontocetus cladistic matrix from this study (TNT).

The raw results of the phylogenetic analyses.

Supplemental Information 4 Notable measurements among odobenines, in mm.

Letters of angles and measures correspond to those indicated in Boisville et al. (2022).

Supplemental Information 5 Comparisons specimens, assigned to Ontocetus emmonsi (A,B) and Odobenus rosmarus (C,D).

From top to bottom: lateral view, occlusal view. Ontocetus emmonsi is represented by USNM PAL 9343 (referred female) (A) and IRSNB M168 (referred male) (B), Odobenus rosmarus is represented by IRSNB 1150B (female) (C) and IRSNB 1150D (male) (D). Scale bar equals 5 cm. Pictures taken and figures drawn by M. Boisville.

We would like to thank Ian James Cruickshanks and Allister Cruickshanks who found the specimen and donated it to the Norwich Castle Museum (NWHCM) of the Norfolk Museums and Archaeology Service (NMAS), Dr. David Waterhouse for providing us with the necessary documentation regarding the specimen, as well as Dr. Anthony G. Irwin (then NWHCM, now NMAS) for allowing access and photographs of the specimen at the time, and Mr. Nigel R. Larkin (then NMAS, now University of Reading) for providing beautiful casts to the Natuurhistorisch Museum Rotterdam (NMR) and the National Museum of Nature and Science (NMNS). We also would like to thank Drs. Pascal Godefroit, Olivier Lambert and Leonard Dewaele for allowing the specimens to be studied at the Institut Royal des Sciences Naturelles de Belgique (IRSNB). We also wish to thank Thierry Hubin (IRSNB) for the photographs of the IRSNB specimens, and Guillaume Duboys de Lavigerie (former IRSNB, ULiège) for his help with the preparation of figures. We would like to thank Mr. Pepijn Kamminga and Ms. Natasja den Ouden for giving us access to the collection of the Rijksmuseum van Geologie en Mineralogie of Leiden, stored in the Naturalis Biodiversity Center. We also wish to thank Prof. Anne Schulp, Prof. Jelle Reumer, Dr. Anne van de Weerd, Dr. Willem Renema, and Mr. Klaas Post for giving us access to Naturalis collections. We would like to thank again Drs. Lambert and Dewaele for their discussion on fossil walruses. We also would like to thank Drs. Sachiko Agematsu and Kohei Tanaka (both Univ. Tsukuba), and Yasunari Shigeta (NMNS/Univ. Tsukuba) for providing useful advice, discussion, and generous encouragement during the course of this study. Finally, we also want to sincerely thank Mr. Klaas Post, curator at the Natuurhistorisch Museum Rotterdam (NMR), for discussing the fossil walruses of the North Sea and for his long-term contributions over the past 30 years for North Sea fossil marine mammals.

Institutional abbreviations

IRSNB Institut Royal des Sciences Naturelles de Belgique, Brussels, Belgium;

NHMUK The Natural History Museum, London, England, U.K.;

NMNS National Museum of Nature and Science, Tsukuba, Japan;

NMR Natuurhistorisch Museum Rotterdam, Rotterdam, The Netherlands;

NWHCM Norwich Castle Museum, Norfolk Museums and Archaeology Service, Norfolk, England, UK;

RGM Rjiksmuseum van Geologie en Mineralogie, Leiden, The Netherlands;

ROM Department of Vertebrate Paleontology, Royal Ontario Museum, Toronto, Canada;

USNM Mammalogy collections, United States National Museum of Natural History, Smithsonian Institution, Washington, D. C., U.S.A.;

USNM PAL Paleobiology collections, United States National Museum of Natural History, Smithsonian Institution, Washington, D. C., U.S.A.

Additional Information and Declarations

Competing Interests

Author Contributions

Data Availability

New Species Registration

The authors declare that they have no competing interests.

Mathieu Boisville conceived and designed the experiments, performed the experiments, analyzed the data, prepared figures and/or tables, authored or reviewed drafts of the article, and approved the final draft.

Narimane Chatar conceived and designed the experiments, performed the experiments, analyzed the data, prepared figures and/or tables, authored or reviewed drafts of the article, and approved the final draft.

Naoki Kohno conceived and designed the experiments, performed the experiments, analyzed the data, prepared figures and/or tables, authored or reviewed drafts of the article, and approved the final draft.

The following information was supplied regarding data availability:

The code and raw data are available in the Supplemental Files.

The following information was supplied regarding the registration of a newly described species:

Publication LSID: urn:lsid:zoobank.org:pub:13F7B996-8009-47B7-A8B6-353D7D14ED81

Ontocetus posti LSID: urn:lsid:zoobank.org:act:14595AD3-E3F5-461C-834E-DF275B95E80D

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
