# Peer review of "New species of Ontocetus (Pinnipedia: Odobenidae) from the Lower Pleistocene of the North Atlantic shows similar feeding adaptation independent to the extant walrus (Odobenus rosmarus)"

_PeerJ, doi:10.7717/peerj.17666_

## Round 0.1 · original submission · Major Revisions

Dear authors,

Thanks for your contribution. I have read the paper as well as the referees' reviews, and I agree with them that the study and material is very interesting and the manuscript is very well written, with a clear guiding thread. In addition, the referees (specially number 3) and I have identified some problems that concern us, and a review of the paper is required previous publication. I detail here the main problems that should be solved:

- A phylogenetic analysis was not carried out, which can help to strength the results obtained in this paper.

- There are too many figures, that really can be merged in one (fig. 2-5).
- See also additional comments of referee 3 about the specific diagnosis of some materials.

In addition, I have identified some minor typesetting errors:
- L. 576: The specific name "posti" is in capital letters.
- L. 605: A point is required following "sexually dimorphic".
- L. 739: Change "mollusc" to "mollusk".
- Figure 6: In the label of Ontocetus, the "sp." is in italics.

Consider all the comments, suggestions, and tips that the referees have made (in their letter or their attached PDFs) to improve the publication, and make the appropriate changes if consider or include an appropiate answer that justify your view and position in the rebuttal letter. Here, I am also attaching a paper that the referee 3 wants to provide to you (https://www.tandfonline.com/doi/full/10.1080/02724634.2023.2296567).

I look forward to seeing the new version of your manuscript.

Thanks for your contribution,
PhD Blanca Moncunill-Solé

·

Basic reporting

The manuscript meets all validity and suitability standards.

Experimental design

No comment.

Validity of the findings

No comment.

Additional comments

Review of PeerJ ms #94587, “New species of Ontocetus (Pinnipedia: Odobenidae) from the Lower Pleistocene of the North Atlantic shows similar feeding adaptation independent to the extant walrus (Odobenus rosmarus)

1. This is an interesting and well-written account of a potentially important fossil odobenine. Given the wide-ranging and somewhat well-studied yet still uncertain taxonomy, phylogeny, and biogeography of odobenids (as the authors make clear), this manuscript could represent a key contribution to the literature. The organization and writing are of high quality, and I have no major or minor recommendations with regard to general or specific language (terms used, etc.).
2. The anatomical/paleontological descriptions of the fossil specimen are appropriately outlined and exceptionally detailed, particularly with regard to muscle attachments. These latter details not only provide valuable paleontological information, but are quite useful in comparing this specimen to the osteology and myology of the extant walrus Odobenus.
3. The literature reviewed here is appropriately broad and up-to-date. The only obvious thing I found to be fixed involves the name of one of the authors cited: Chris Marshall, whose name is correctly spelled throughout the References section and in some places where cited in the text, although the name is misspelled (as Marschall) in at least three places, in lines 489 and 500.
4. With regard to the writing, the main suggestion I can offer is that some of the subsections of the Discussion have very long paragraphs that would benefit from being broken into shorter paragraphs. In some places there are obvious shifts in content where breaks could be introduced.
5. The figures are very nice, with crisp photographs and drawings and nice use of color. These figures do what the best figures do: not merely reiterate material within the text, but truly add another dimension of presented information.
6. I noted that to reduce confusion between the genera Ontocetus and Odobenus, the former is often abbreviated as On. Perhaps spelling out the full name of the genus would be a safer bet to avoid any possible confusion.
7. I like the breakdown of the Discussion into varied sections covering such topics as sexual dimorphism and paleoecology (the latter including nice material on Ontocetus dispersal and distribution, particularly with regard to climate change and consequences for prey fauna). On this note, I agree that the Central American Seaway likely provided a more logical route for dispersal than through the Arctic, but I wonder if the authors could comment on the potential for odobenines to have spread through the Arctic (and the role of climate change in affecting walrus biogeography, not only in the Pliocene but also much more recently).
8. The morphological/dental features linked to suction feeding, and particularly to strong suction-based molluscivory (e.g., palatal vaulting and a very heavy, fused mandibular symphysis) are nicely explained. I wonder what the authors think about the oft-speculated role of the C1 tusks in odobenines (whether for “tooth walking” and hauling out of water, “digging” in ice or other substrates, or male-male combat and/or display, and so on). I am particularly interested in another potential role of these tusks in facilitating suction feeding on benthic molluscs, as suggested by de Muizon and Domning in their accounts of Odobenocetops—namely, as providing a “sled” (like the runners below ROVs) that maintains a useful distance between the gape and the benthic substrate, so that clams or other molluscs can be more efficiently sucked up and harvested from the seafloor (or sucked right out of their shells, as some people have proposed). Given the very large tusks of Ontocetus (and Valenictes), even if these are smaller than in extant walruses, I wonder if the authors agree that these teeth might provide some possible benefit in foraging on benthic molluscs (even though this hypothesis remains speculative, although the striking convergence between the tusks of Odobenocetops and Odobenus cries out for some functional explanation, even a “just so” speculative one).
9. Regarding potential dental function, I wonder if the authors can say (or show) more about the tooth wear briefly mentioned in lines 379-381.
10. Also, it is my understanding that dentition (e.g., tooth counts) is somewhat variable in living walruses, unlike most mammals, where it is highly uniform. Does this have any bearing on this fossil?
11. The section on sexual dimorphism is thorough and presented well. I wonder how much this would affect the diet and foraging of male vs. female Ontocetus (especially Ontocetus posti), or are they presumed to have fed similarly?
12. On a related note, what about the age of the fossil, not in geological terms but in development of the animal, which is clearly noted to be an adult. Are there any indications (for example, wear patterns) that it was a particularly old or young adult? Is the basic ontogenetic timing of this species presumed to be similar to that of Odobenus rosmarus?
13. Overall, this is a nicely organized and well-written manuscript, full of ideas and supporting evidence.

Reviewer 2 ·

Basic reporting

I am delighted with the quality of this study. The text is well-written and logically organized. The introduction provides an up-to-date overview of Pliocene and Pleistocene walruses, identifies the major taxonomic controversies, and contextualizes the reassessment of the mandibles conducted here. Likewise, the morphological descriptions and comparisons are adequate and the figures are relevant and useful.

Experimental design

The manuscript provides all necessary information about the fossil specimens, including their stratigraphic provenance and collection numbers. The materials are correctly figured and raw measurements are presented.

Validity of the findings

The morphological variability exhibited by NWHCM 1996.1 and the referenced specimens (i.e., IRSNB M156 & RGM.St 119589) relative to Odobenus rosmarus and Ontocetus emmonsi is apparent and supports the recognition of a new species. However, given the wide range of morphological intra- and interspecific morphological variability within pinnipeds (and odobenids), including their mandibles, a quantitative exploration of morphological variability may be beneficial. I am not suggesting that the lack of a quantitative approach invalidates the results of this study. Rather, a more thorough exploration of morphological variability is likely to strengthen the current results. If principal component analysis is not feasible (due to the fragmentary nature of some of the fossil specimens), bivariate plots of measurements associated with diagnostic characters (e.g., transverse vs. anteroposterior canine diameter, tooth row length vs. canine APD, symphysis APD vs. transverse distance, etc.) including a larger sample size of extant walrus species and fossil taxa may also have conclusive results.

Additional comments

Dear Editor
Thank you for the opportunity to review the manuscript titled “New species of Ontocetus (Pinnipedia: Odobenidae) from the Lower Pleistocene of the North Atlantic shows similar feeding adaptation independent to the extant walrus (Odobenus rosmarus).” This manuscript reevaluates three mandible specimens of Odobenidae from the eastern North Atlantic, resulting in the establishment of a new Ontocetus species. This is a good paper, with adequate morphological descriptions, comparisons, and figures. Although it does not invalidate the current results, I believe this study would benefit by incorporating a more quantitative comparison of the morphological variability observed between specimens of Odobenus rosmarus, Ontocetus emmonsi, and Ontocetus posti. Moreover, I suggest that supplementary figures 1 and 2 should be moved to the main text, as the specimens and morphological comparisons there are relevant to the main conclusions. If the number of figures bothers you, you might want to consider combining figures 8 and 9 into a single figure with multiple panels. Likewise, I suggest moving Table S1 to the main document too.

Overall, I suggest a very minor revision. If the authors would like to consider my recommendations, I think this could make a solid manuscript an even stronger paper.

Miscellaneous comments:
Lines 49-50. This sentence should be edited. Walruses are one of the largest…
Line 56-58. Please add a reference.
Line 85. Please replace “or” by “and”
Line 149. Please remove the dot before the “:”
Lines 202-203. Could you quantify how much larger is the lower canine relative to postcaines?
Line 205. What do you mean by “its relative size?” Please be more precise, and, if possible, quantify it.
Lines 206-207. What do you mean by “short distance between the dorsal margin and coronoid process” Please clarify.
Lines 209-210. Please quantify how much shorter the mandibular symphysis is relative to the mandible's length.
Lines 210-211. I suggest rephrasing this sentence to “...a horizontal ramus with a well-curved lateral occlusal outline, that accounts for the shortening of the rostrum.”
Line 301. Instead of “broken away” you could say that the anterior tip of the symphyseal region is missing or not preserved.
Line 303. Do you mean conical? (instead of clinical)
Line 303. Perhaps: “forming a small and smooth tubercle”
Lines 336-337. How much short? Please quantify it.
Lines 408-409. Please expand on the assignation of IRSNB M168 as Ontocetus emmonsi. Was this specimen assigned by the authors or by someone else?
Line 461. Please add a reference.

·

Basic reporting

This manuscript by Boisville et al. is a competent addition to the study of fossil walruses, reporting several early Pleistocene aged mandibles from the North Sea (UK, Netherlands) that represent a new species, named herein as Ontocetus posti. The fossils are well-described and well-figured, and the fossils themselves are quite interesting and represent an unusual time period that is otherwise very poorly sampled. In 2018 we reported a tusk that is slightly younger from South Carolina. However, there are a number of major and minor issues that need to be addressed, and for the time being, I recommend major revision. Note that I do not necessarily endorse all of these statements below – I found the reasoning behind the use of Ontocetus emmonsi compelling when Kohno and Ray (2008) proposed it, but have always felt that mandibular remains are preferable to isolated tusks as far as designating diagnoseable taxa is concerned – and have always been more sympathetic to the use of Alachtherium cretsii.
Additionally, I have a new paper in press that is about to come out in Journal of Vertebrate Paleontology reporting a new species of Valenictus from California. I’ve attached a copy of the page proofs* – this study includes a considerably updated phylogenetic matrix for Odobenidae and, given that I request a phylogenetic analysis below, would mean that the authors will not have to reinvent the wheel, so to speak. There are also some comments on Plio-Pleistocene occurrences of Ontocetus and Odobenus that will be relevant.
All minor comments are present in the marked up pdf. This manuscript is an excellent start – but needs a little more thought. If the authors can address all of these concerns it will be a fantastic and thought-provoking contribution.
Sincerely,
Robert W. Boessenecker, Ph.D.
Charleston Center for Paleontology, Charleston, South Carolina, USA

*I attempted to attach the page proofs in a zip file with the marked up review pdf, but I can only upload a single pdf. Since this review is signed, I can either 1) email the pdf to the editor or 2) email it directly to the lead author (Boisville) but will wait to do so until the editor contacts me with instructions.

Major comments
1) Genus level assignment. In many ways I think this mandible is morphologically closer to Odobenus than to Ontocetus, especially in the shortened mandibular symphysis. Many of the features listed in the diagnosis as supposedly precluding assignment to Odobenus (enlarged lower canine, presence of incisors, slanted anterior margin, high coronoid process) are plesiomorphic features absent in Odobenus rosmarus but in actuality could very well be expected to occur in an early species of Odobenus. In fact, one of these – retention of incisors – is already known in an earliest Pleistocene skull of Odobenus reported by Miyazaki et al. 1992. The similarities of the mandibles with Odobenus are perhaps most striking in dorsal view. Given the similarities between both genera, I’m actually quite surprised that some sort of morphometric or phylogenetic analysis was not conducted. A phylogenetic analysis should be conducted to test assignment of this species to one genus or the other.
2) On that note, I’m further surprised that the authors did not consider, given the transitional morphology of this specimen, a derivation of Odobenus directly from Ontocetus. In the past I have proposed (Boessenecker et al. 2018) that the two taxa were separated in time – but this very well may be caused by sampling bias and the lack of good early Pleistocene marine mammal sites in the North Atlantic (and beyond). Serious thought should be given to an ancestor-descendant relationship. Even if the authors do not think this is plausible, it needs to be borne out in this study. This will weigh heavily on the genus level assignment (see point 1 above). If some sort of ancestor-descendant relationship was supported, that would be extremely interesting.
3) There’s a little bit of a philosophical inconsistency with prior work here regarding the diagnoseability of Ontocetus emmonsi v. Odobenus/Trichecodon koninckii. Kohno and Ray (2008) considered all Pliocene odobenines from the North Atlantic one species with the oldest available name Ontocetus emmonsi, synonymizing Alachtherium cretsii – based on a highly diagnostic mandible – with this species, the holotype of which is a fragmentary tusk. The authors of this study contend that the fragmentary tusk holotype of O/T koninckii, which is only slightly less completely preserved than the O. emmonsi holotype, is non-diagnostic. Should these taxa be based on well-preserved mandibles, or tusk fragments? If the former, then perhaps Alachtherium cretsii should be resurrected. If the latter, then perhaps this new species is simply best identified and redefined as Ontocetus/Odobenus koninckii.
4) Additionally, the discovery that at least some Pliocene-Pleistocene walruses are NOT assignable to Ontocetus emmonsi perhaps erodes the taxonomic framework of Kohno and Ray (2008) supporting the use of O. emmonsi as the oldest available and readily diagnoseable taxon/name – since many, many specimens consist of isolated tusks and postcrania and cannot be directly compared with either the A. cretsii holotype mandible and similar specimens or the O. posti mandible morphotype. Should we still be using the genus Ontocetus, or instead Alachtherium?

Moderate comments

1) The diagnosis, and the comparisons section, need to be more quantitative with some proportions expressed as % or as ratios.
2) All supplementary figures should be included as main figures within the text of the paper; perhaps the figures of referred specimens could be combined into a single composite figure.
3) The language is generally quite good throughout but occasionally there are some odd word choices; for most of these I've provided a suggested correction.


Figure comments

1) Figure 6: “enamels” should be singular (enamel); “specialization” should be with a Z since PeerJ uses American English (rather than British English)
2) Figure S2-1: prominent is misspelled

Experimental design

see above

Validity of the findings

see above

---

## Round 0.2 · Minor Revisions

Dear authors,

Thanks for this new version of your paper. I have read it and most of the previous comments of reviewers have been taken into consideration. This makes the study more solid and robust. Congratulations for it. I have read the new comments of the referees and I agree with them that some minor changes (most of them of typesetting or spelling errors) should be addressed previous acceptance. Please, consider also the comments of referee 3 who pointed out that some aspects of the new results are not discussed, and it would be better if you can include some paragraphs about them in a new version.

Besides, referee 3 also indicated to me that figure 5 of your paper is based on his illustrations published in a previous study. Even in the case that it is not exactly the same figure, but the figure that he elaborated is used as a model and your figure 5 is based on it, please, indicate him into the credits in order to not have problems with author credits. Take care about the comment of him in this figure (maybe you can write to him asking permission). This is very important to solve this problem previous to publication.

Thanks for your time and work.

Hoping to see your new version of this paper very soon.

Blanca

·

Basic reporting

This revised resubmission is considerably improved in terms of both content and organization. The figures are simplified, clear, and easy to understand. The methods and results are described in greater detail than in the original submission, and the description of the actual fossil itself is more detailed.

Experimental design

No comment

Validity of the findings

There is now much additional information concerning morphometry/morphology and paleoecology, including the chronology of dispersal and biogeography. Perhaps most importantly, the phylogenetic placement of this fossil is now considered more carefully and reliably. Even if the final species-level assignment is indeterminate, the genus-level assignment (whether Ontocetus or Odobenus) is handled thoroughly.

Additional comments

There may still be questions surrounding this fossil, but this manuscript does a nice job discussing all the relevant issues. I am satisfied that the authors adequately addressed my stated concerns and those of the other reviewers, and I thank them for improving their manuscript accordingly.

Reviewer 2 ·

Basic reporting

The text is logically structured and contains appropriate background information. It provides reasonable evidence to support this paper's findings.

Experimental design

No comment

Validity of the findings

No comment

Additional comments

Dear authors
I am pleased that the authors have considered my and other reviewers' comments. The manuscript reads better now and presents more solid evidence supporting the identification of the new species, laying out more clearly its implications for walruses' evolution and paleoecology in the North Atlantic. I only have a few minor comments that should be addressed before publication (see below).

Lines 115-120. This sentence is too long. I suggest the following edit: "The present study focuses on re-evaluating a pair of mandibles from the Lower Pliocene of Norwich, Great Britain, and a mandible from the Upper Pliocene of Antwerp, Belgium. At least the latter was historically reported under different taxonomic names (Trichecodon koninckii by van Beneden, 1877; Odobenus koninckii by Deméré, 1994b), but these names have been discarded as nomen nodum or nomen dubium (e.g., Rutten, 1907; van der Feen, 1968; Kohno and Ray, 2008)."

Line 138. The correct acronym for the National Museum of Natural History paleobiology collection is USNM PAL.

Line 179. Following the prior comment, the correct collection number for the female Ontocetus emmonsi specimen here is USNM PAL 374273 and elsewhere. Please see https://collections.nmnh.si.edu/search/paleo/?ark=ark:/65665/3484c587429c5451fab12ad1ef2900b44 for reference.

Lines 196, 205, etc. There is a spelling in Boessenecker et al. (2024).

Line 229. Add a period after the reference.

Line 251. "Odobenus" should be italicized.

Lines 261, 360. Please correct the spelling in Deméré 1994.

Lines 717, 719, 901, 905. Please correct the acronym for USNM V 475482, 9343, etc.

Figures 3 and 4. These two figures could be merged into a single plate.

Figure 7. Please add the collection numbers of the specimens included in the cluster dendrogram. This figure might also be merged with Figure 8 in a sole plate.

·

Basic reporting

Dear Editor,
I find the manuscript by Boisville et al. greatly improved, but it still needs some work. The phylogenetic analysis is quite interesting and is a valuable inclusion to the paper – but has a very bare-bones description of the results (three sentences) and does not appear at all in the discussion (and, neither do the morphometric results). The updated diagnoses are good, but there still is very little in the discussion about the generic assignment of O. posti and whether it belongs in Ontocetus or Odobenus. The generic assignment is perhaps somewhat clarified thanks to the diagnoses, but needs to be highlighted in the discussion – likely as a new discussion paragraph/heading, at the beginning of the discussion. The morphological arguments for assignment to Ontocetus and Odobenus should be outlined, the morphometric results, and the phylogenetic results. The morphometric results have an intercalation of Ontocetus emmonsi and Ontocetus posti specimens – supporting the author’s interpretation – but the phylogenetic analysis, on the other hand, finds O. posti to be sister to the Valenictus + Odobenus clade. This can be interpreted in two or more ways: 1) Ontocetus is paraphyletic, with O. posti being the geochronologically youngest of these species, or 2) O. posti is something else, more closely related to Odobenus, and 3) possibly ancestral to Odobenus. One major problem is that we have late Pliocene and early Pleistocene specimens of Odobenus from Japan, but none of the published specimens are mandibles – and as I highlighted in my first review, the O. posti holotype more or less is precisely what an ancestral Odobenus mandible would look like. Now, it’s fine if the authors don’t take these ideas seriously – but they are, at present, reasonable alternative hypotheses that, form my perspective, are just as likely given the available data. Given the incompleteness of the specimen (isolated mandible without a skull), the authors need to have a section 1) outlining and defending the genus-level allocation and 2) discussing and eliminating alternative hypotheses – because, like it or not, the last two sections of the discussion hinge on the genus that posti is assigned to. In the present draft, the genus level allocation is presented as a fact to be accepted and not as a hypothesis.
I do not find the arguments for a sea lion-like raptorial feeding strategy compelling, given that Ontocetus has 1) large tusks, 2) peg-like teeth, and 3) a vaulted palate – all of this suggests an Odobenus-like feeding behavior. This section should be reduced as I think the interpretation is incorrect.
Kind regards,
Robert W. Boessenecker

Experimental design

See above.

Validity of the findings

See above.

Additional comments

Line 175: Systematic…what? I think there’s a word missing.
204: supplemental appears twice.
235: Boessenecker is misspelled here and throughout – surprisingly, there are four different ‘e’s in the name! Don’t forget the one after the ‘ss’. Likewise, it seems as though this missing ‘e’ has been accidentally given to the similarly named (but unrelated) Mark Bosselaers (who only has two ‘e’s)!
244-252: I like the inclusion of the new characters, however, the rationale for the other character changes is not explained. Why were polymorphic states removed, and why were some characters changed from ordered to unordered?
283: I believe “Amended” is the proper word here (after some argumentation with Dr. A. Poust)
684-690: this is a bit skimpy; how does the topology compare with Boessenecker et al. 2024? Are there any changes to bootstrap support? What is the bootstrap support, anyway? There’s no mention of some very fundamental differences in topology for the ‘imagotariine’ walruses, which seem instead to plot out as early members of a desmatophocid-otariid-phocid clade.
956—957: I disagree that an Arctic dispersal would have been a problem; the Arctic was considerably warmer during the Pliocene and during Pleistocene interglacial periods (MIS 11, for example, had sea otters inhabiting the Arctic) and Phoca vitulina crossed the arctic during the Pleistocene; warm equatorial waters seem to be more of a barrier for marine mammal dispersal.
961-973: A very long run-on sentence- please break up. Furthermore, the presence of Herpetocetus along all four coastlines of the north Pacific and north Atlantic just as strongly argues for a trans-Arctic dispersal. I’m not sure what the relevance is here – though it’s certainly an interesting coincidence. On the other hand, the presence of Herpetocetus in all four regions makes it a better candidate for a CAS dispersal – unlike Ontocetus, it’s been found on the Pacific coast. In fact, this makes Herpetocetus a rather poor point of comparison because it highlights the difference in biogeographic distribution. The fossil record of Eschrichtius is actually quite similar (WNP, ENP, WNA, ENA) to Herpetocetus, and like Ontocetus, are known/inferred benthic suction feeders without evidence of a trans-equatorial distribution to the southern hemisphere.
Regardless, this central American hypothesis still suffers from a lack of fossils of Ontocetus from the entire Pacific coast of North America – and this caveat should be explained more obviously in the text.
964: There aren’t many Messinian deposits on the east coast that are rich in marine vertebrates outside Florida – so I suspect the oldest occurrence being in Florida is just happenstance. There’s only really one other latest Miocene deposit, the Eastover Formation, and only a handful of cetacean specimens have been collected from it.
974-5: Also, San Diego Formation, California.
Figure 3: Looks like the two original figures were stuck together quickly without consideration of font size and possible improvements to layout. The labels can be moved around (and should be of equal font size) and the photographs can be enlarged somewhat – at present, there is a lot of wasted white space. These photos should take up as much room as possible – and can be enlarged by 40-50%.
Figure 4: ditto.
Figure 5: needs credit to the original artist (myself). Re-drawing the images, with or without permission, requires credit. These look identical to the original illustrations (as they should), but credit for the originals is needed 100% of the time. E.g. “redrawn from Boessenecker by N. Chatar” or something similar. [I have double checked my email, and I do not have an email anywhere in my inbox about a permission form; please re-send].
Figure 6: see comments for figures 3-4.
Figure 9: Where are the bootstrap values? These should be labeled on this cladogram.
Figure 10: what’s the source of the geochronologic ranges?
Figure 11: why not put the lateral side on a column on the left, with posterior view on the right, to make the figure larger and therefore higher resolution?

---

## Round 0.3 · accepted · Accept

Dear authors,

Congratulations! This final version has addressed all the referees' comments succesfully. From my point of view, the reviewing process has been very important, and has improved the quality of your paper and strengthened the results. A very clear guiding thread is present, and all the sections are very well explained. Thanks for your time and patience in the reviewing process.

I have made a last reading, and I have noted some minor typographical errors. I detail them below. Please, correct them in the fully typeset publication proofs, together with others that you can identify in them, before final publication.

I wish you all the best in your research projects and carreer.
Cheers,

PhD Blanca Moncunill-Solé

Minor typographical changes:
line 72 and 75: upper or Upper Miocene? In some place in capital letters in other lower case.
line 97: inclue "and" before "A. africanum".
line 120: "nomen nodum" and "nomen dubium" in italics.
line 137: include a point following "U.K", before ";".
line 212: delete the space between "0=" adn "same level".
line 212: change "." to "," after "anterior to p2".
line 220: change "(Present : 0, Absent : 1)" to "(Present=0, Absent=1)"
line 221: delete the spaces before and after "=" when describing the states of "character 145" (not in capital letters).
line 290: change "check" to "cheek".
line 342 and line 356: the dash that connects the two ages sometimes is an em dash, and others an en dash. Check all the paper, and be consistent.
line 368: "nomen nudum" in italics.
line 373: "nomen dubium" in italics.
line 388: change "49 mm" to "49.0 mm".
line 465: include a space between "view" and "(Fig. 3)".
line 522: delete the space between "RGM." and "St.119589".
line 544: sometimes is "NMR7472", sometimes "NMR 7472". Please check all ms, and be consistent.
line 605: change "Alghough" to "Although".
line 731: "temporalis" in italics.
line 753 and line 754: sometimes there is a dash between "suction" and "feeding", but there is not in other places. Please, check all paper and be consistent.
Caption of Fig. 5: Include a space between "10" and "cm" when describing the scale bar.
Figure 10: The muscles names should be in italics.
Caption of Fig. 11: Include a space betwen "1.7 Ma)." and "Ontocetus".